# Diversification dynamics in the Neotropics through time, clades, and biogeographic regions

Andrea S Meseguer[1,2]*, Alice Michel[1,3], Pierre-Henri Fabre[1,4,5], Oscar A Pérez Escobar[6], Guillaume Chomicki[7], Ricarda Riina[2], Alexandre Antonelli[6,8,9], Pierre-Olivier Antoine[1], Frédéric Delsuc[1], Fabien L Condamine[1]

[1]Institut des Sciences de l'Evolution de Montpellier (Université de Montpellier | CNRS | IRD | EPHE), Place Eugène Bataillon, Montpellier, France; [2]Real Jardín Botánico (RJB), CSIC, Madrid, Spain; [3]Department of Anthropology, University of California, Davis, United States; [4]Mammal Section, The Natural History Museum, Cromwell Road, London, United Kingdom; [5]Institut Universitaire de France (IUF), Paris, France; [6]Royal Botanic Gardens, Kew, United Kingdom; [7]Ecology and Evolutionary Biology, University of Sheffield, Sheffield, United Kingdom; [8]Gothenburg Global Biodiversity Centre, Department of Biological and Environmental Sciences, University of Gothenburg, Gothenburg, Sweden; [9]Department of Plant Sciences, University of Oxford, Oxford, United Kingdom

*For correspondence:
asanchezmeseguer@gmail.com

Competing interest: The authors declare that no competing interests exist.

**Abstract** The origins and evolution of the outstanding Neotropical biodiversity are a matter of intense debate. A comprehensive understanding is hindered by the lack of deep-time comparative data across wide phylogenetic and ecological contexts. Here, we quantify the prevailing diversification trajectories and drivers of Neotropical diversification in a sample of 150 phylogenies (12,512 species) of seed plants and tetrapods, and assess their variation across Neotropical regions and taxa. Analyses indicate that Neotropical diversity has mostly expanded through time (70% of the clades), while scenarios of saturated and declining diversity account for 21% and 9% of Neotropical diversity, respectively. Five biogeographic areas are identified as distinctive units of long-term Neotropical evolution, including Pan-Amazonia, the Dry Diagonal, and Bahama-Antilles. Diversification dynamics do not differ across these areas, suggesting no geographic structure in long-term Neotropical diversification. In contrast, diversification dynamics differ across taxa: plant diversity mostly expanded through time (88%), while a substantial fraction (43%) of tetrapod diversity accumulated at a slower pace or declined towards the present. These opposite evolutionary patterns may reflect different capacities for plants and tetrapods to cope with past climate changes.

## Editor's evaluation

This important work by Meseguer et al. depicts findings that substantially advance our understanding of clade diversification across major Neotropical bioregions. The evidence that summarises the evolutionary diversity dynamics of 150 time-calibrated clades of neotropical plants and animals data is convincingly presented with current state-of-the-art analyses. The work will be of interest to evolutionary biologists and biogeographers working to understand the origins of the most biodiverse land mass on the planet.

## Introduction

Comprising most of South America, Central America, tropical Mexico, and the Caribbean Islands, the Neotropics are the most biodiverse region on Earth, home to at least a third of global biodiversity (*Raven et al., 2020*). This region not only includes the largest tropical rainforest, Amazonia, but also 8 of the world's 34 biodiversity hotspots (*Mittermeier et al., 2011*). The tropical Andes, in particular, are considered to be the most species-rich region in the world for amphibians, birds, and plants (*Myers et al., 2000*), while Mesoamerica and the Caribbean Islands are the richest regions for squamates, and Amazonia has been identified as the primary biogeographic source of Neotropical biodiversity (*Antonelli et al., 2018c*). The drivers underlying the origins and maintenance of the extraordinary biodiversity of the Neotropics are hotly debated in evolutionary ecology and remain elusive (*Gentry, 1982*; *Simpson, 1980*; *Antonelli and Sanmartín, 2011a*; *Hoorn et al., 2010*; *Rull, 2011*; *Antonelli et al., 2018a*).

Attempts to explain Neotropical diversity traditionally relied on two evolutionary models. In the first, tropical regions are described as the '*cradle of diversity*', the centre of origin from which species appeared, radiated, and colonized other areas (*Diels, 1908*; *Bews, 1927*; *Ingvar et al., 1968*). In the other, tropical regions are considered a '*museum of diversity*', where species suffered relatively fewer environmental disturbances over evolutionary time, allowing ancient lineages to be preserved for millennia (*Simpson, 1980*; *Stebbins, 1974*; *Wallace, 1878*). Although not mutually exclusive (*McKenna and Farrell, 2006*), the cradle vs. museum hypotheses primarily assume evolutionary scenarios in which diversity expands through time without limits (*Hey, 1992*). However, expanding diversity models may be limited in their ability to explain the entirety of the diversification phenomenon in the Neotropics; for example, expanding diversity models cannot explain the occurrence of ancient and species-poor lineages in the Neotropics (*Condamine et al., 2015*; *Antonelli and Sanmartín, 2011b*; *Gibb et al., 2016*) or the decline of diversity observed in the Neotropical fossil record (*Hoorn et al., 1995*; *Jaramillo et al., 2006*; *Antoine et al., 2017*). Although the concepts of cradle and museum have contributed to stimulate numerous macroevolutionary studies, a major interest is now focused on the evolutionary processes at play rather than the diversity patterns themselves (*Vasconcelos and O'Meara, 2022*). Four alternative evolutionary trajectories of diversity dynamics could be hypothesized to explain the accumulation of Neotropical diversity observed today (*Figure 1*):

### Gradual expansions (Scenario 1)

This scenario proposes that species richness accumulated gradually through time in the Neotropics until the present, due, for example, to constant speciation and extinction rates. The gradual increase model received substantial support in the early and recent literature (*Wallace, 1878*; *Couvreur et al., 2011*; *Derryberry et al., 2011*; *Santos et al., 2009*; *Schley et al., 2018*; *Harvey et al., 2020*), and is generally associated with the long-term environmental stability and large extension of the tropical biome across the South American continent (*Simpson, 1980*; *Stebbins, 1974*).

### Exponential expansions (Scenario 2)

An exponential increase in diversity model asserts that species richness accumulated faster towards the present. Such a pattern can result, for example, from constant extinction and increasing speciation rates, or constant speciation and decreasing extinction. Support for this model generally comes from studies suggesting that recent geological and climatic perturbations, mostly associated with the elevation of the Andes, promoted increases of diversification (*Hoorn et al., 2010*; *Rull, 2011*; *Antonelli et al., 2018b*). This diversity scenario is probably the most supported across Neotropical studies, although never quantified, with models of increasing speciation (*Haffer, 1969*; *Richardson et al., 2001*; *Meseguer et al., 2020*; *Erkens et al., 2007*; *Hughes and Eastwood, 2006*; *Esquerré et al., 2019*; *Drummond et al., 2012*; *Lagomarsino et al., 2016*; *Pérez-Escobar et al., 2017*; *Musher et al., 2019*; *Olave et al., 2020*) more often put forward than models of decreasing extinction (*Antonelli and Sanmartín, 2011b*).

### Saturated or asymptotic expansions (Scenario 3)

A saturated diversity model postulates that species richness accumulated more slowly towards the present than in the past, reaching a diversity plateau. This can result from constant extinction and decreasing speciation, for example, such that speciation and extinction rates become equal towards

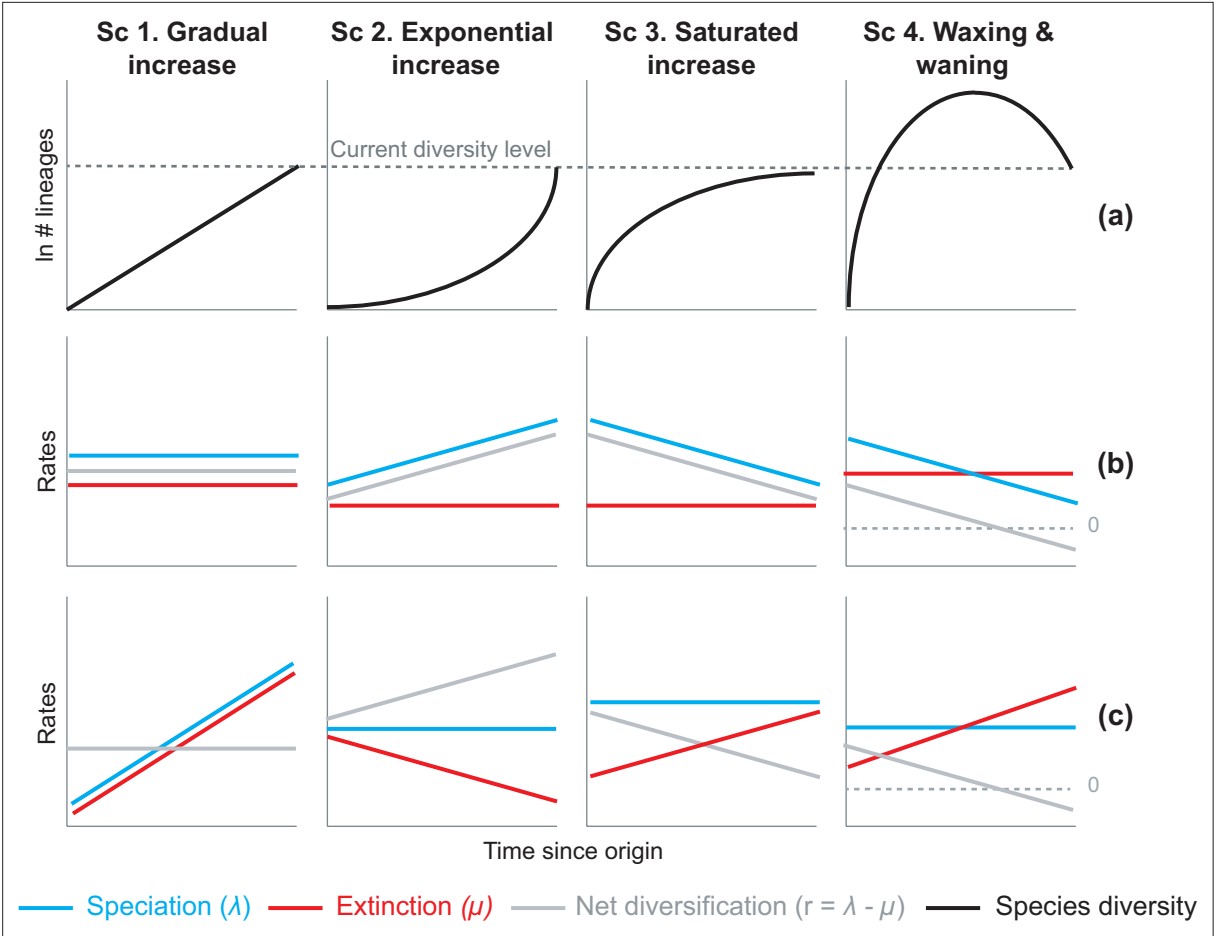

**Figure 1.** Alternative hypotheses to explain current Neotropical diversity. (**a**) Main species richness dynamics through time, and (**b,c**) the alternative evolutionary processes that could generate the corresponding patterns. (Sc. 1a) A gradual increase of species richness could result from constant speciation and extinction rates (1b), or through a comparable increase in speciation and extinction rates (1c). (Sc. 2a) An exponential increase in species numbers could be attained through constant extinction and increasing speciation (2b), or constant speciation and decreasing extinction rates (2c). (Sc. 3a) Saturated increase scenarios, with species accumulation rates slowing down towards the present, could result from constant extinction and decreasing speciation (3b), or through constant speciation and increasing extinction rates (3c). (Sc. 4a) Waxing and waning dynamics could result from constant extinction and decreasing speciation (4b), or constant speciation and increasing extinction (4c). Waxing and waning scenarios differ from saturated increases in that extinction exceeds speciation towards the present, such that diversification goes below 0. Scenarios (**b–c**) represent the simplest and most general models to explain species richness patterns in (**a**), but other combinations of speciation and extinction rates could potentially generate these patterns; for example, an exponential increase of species (2a) could also result from increasing speciation and punctual increases in extinction, or through increasing speciation and decreasing extinction.

the present. Diversification decreases could be due to ecological limits (*Rabosky, 2009*), damped increases (*Cornell, 2013*; *Morlon et al., 2010*), or abiotic fluctuations (*Condamine et al., 2019a*). Some studies support this model for the Neotropics, and they generally associate it with an early burst of diversification under favourable climatic conditions, followed by decelerations due to global cooling, and dispersal constraints (*Santos et al., 2009*; *Phillimore and Price, 2008*; *Fine et al., 2014*; *Cadena, 2007*; *Weir, 2006*).

## Declines in diversity (Scenario 4)

Waxing and waning dynamics characterize clades that decline in diversity after periods of expansion. In a declining dynamic, diversification rates also decrease towards the present, but differ from saturated diversity in that extinction exceeds speciation, and diversity is lost. Waxing and waning dynamics may seem unlikely in a tropical context, but evidence for tropical diversity declines has been found at the global scale (*Meseguer and Condamine, 2020*; *Quental and Marshall, 2013*; *Foote et al., 2007*) and at the Neotropical scale in the fossil record (*Hoorn et al., 1995*; *Jaramillo et al., 2006*; *Antoine*

*et al., 2017*; *Archibald et al., 2010*; *Salas-Gismondi et al., 2015*; *Jansa et al., 2014*; *Carrillo et al., 2020*). Fossil studies additionally suggest a link between decreases in Neotropical diversity and global temperature. For example, plant diversity inferred from fossil morphotypes reached its maximum levels during hyperthermal periods in the Eocene, and decreased sharply with subsequent cooling (*Hoorn et al., 1995*; *Jaramillo et al., 2006*; *Wilf et al., 2005*).

Despite an increasing number of evolutionary studies on Neotropical groups, today the prevalence of these alternative modes of species accumulation and diversification (*Figure 1*) at a continental scale has been difficult to tease apart empirically (Question 1). Yet, such an assessment would contribute to understand the origin and maintenance of Neotropical diversity. Illuminating the historical causes of Neotropical diversity further requires a closer look at the regional determinants of diversification. Are diversity trends (Sc. 1–4) related to specific environmental drivers (Question 2), geographic settings (Question 3), or taxonomic groups (Question 4) in the Neotropics?

Previous studies indicate that diversification rates might be structured geographically in the Neotropics (*Harvey et al., 2020*; *Jetz et al., 2012*; *Quintero and Jetz, 2018*; *Rangel et al., 2018*), with geography and climate being strong predictors of evolutionary rate variation (*Quintero and Jetz, 2018*; *Rangel et al., 2018*). For example, speciation may be high in regions subject to environmental perturbations, such as orogenic activity (*Esquerré et al., 2019*; *Lagomarsino et al., 2016*; *Vasconcelos et al., 2020*; *Pouchon et al., 2018*; *Madriñán et al., 2013*), and often not associated with current species richness (*Harvey et al., 2020*; *Quintero and Jetz, 2018*). Still, little is known on the geographic structure of long-term Neotropical diversification. Studies investigating spatial patterns of Neotropical diversification focus on long-term diversification dynamics of particular clades, for example, diversification trends of orchids across Neotropical regions (*Pérez-Escobar et al., 2017*), or cross-taxonomic patterns in shallow evolutionary time, that is, present-day speciation rates (*Harvey et al., 2020*; *Quintero and Jetz, 2018*; *Smith et al., 2014*). However, present-day speciation rates might not represent long-term diversification dynamics, especially when rates vary through time. Present diversification could be higher in one region than another without providing information on the underlying trend in diversification. Under time-variable rate scenarios, analysing diversity trends is crucial, but requires changing the focus from species to clades as units of the analyses. Unfortunately, there is still a lack of large-scale comparative data across wide phylogenetic and ecological contexts (*Vasconcelos et al., 2020*; *Eiserhardt et al., 2017*). Given the long history and vast heterogeneity of the Neotropics, general insights can only be provided if long-term patterns and drivers of diversification are shared among Neotropical lineages and areas.

This lack of knowledge may be also due to the challenge of differentiating between evolutionary scenarios based on birth-death models and phylogenies of extant species alone (*Nee et al., 1994*; *Rabosky, 2010*). Recent studies have raised concerns on difficulties in identifying parameter values when working with birth-death models under rate variation scenarios (*Stadler, 2013*; *Burin et al., 2019*), showing that speciation (birth, $\lambda$) and extinction (death, $\mu$) rates sometimes cannot be inferred from molecular phylogenies (*Louca and Pennell, 2020*). This calls for (i) analysing 'congruent' models with potentially markedly different diversification dynamics but equal likelihood for any empirical tree (*Louca and Pennell, 2020*), or (ii) implementing a solid hypothesis-driven approach, in which a small number of alternative hypotheses about the underlying mechanism are compared against data (*Morlon et al., 2022*).

Based on an unparalleled comparative phylogenetic dataset containing 150 well-sampled species-level molecular phylogenies and 12,512 extant species, we evaluate the prevalence of macroevolutionary scenarios 1–4 (*Figure 1*) as general explanations for Neotropical diversification at a continental scale (Q1), their drivers (Q2), and their variation across biogeographic units (Q3) and taxonomies (Q4). To address Q3, we previously identify long-term evolutionary arenas of Neotropical diversification suitable for comparison. Depending on the taxonomic source (*Raven et al., 2020*; *Meseguer et al., 2020*), our dataset represents ~47–60% of all described Neotropical tetrapods, and ~5–7% of the known Neotropical plant diversity.

# Results

## Neotropical phylogenetic dataset

We constructed a dataset of 150 time-calibrated clades of Neotropical tetrapods and plants derived from densely sampled molecular phylogenies (*Figure 2*; *Figure 2—source data 1*; *Meseguer, 2021*). The dataset includes a total of 12,512 species, consisting of 6222 species of plants, including gymnosperms and angiosperms (66 clades, representing 5–7% of the described Neotropical seed plants); 922 mammal species (12 clades, 51–77% of the Neotropical mammals); 2216 bird species (32 clades, 47–59% of the Neotropical birds); 1148 squamate species (24 clades, 30–33% of the Neotropical squamates); and 2004 amphibian species (16 clades, 58–69% of the Neotropical amphibian diversity). Each clade in our dataset includes 7–789 species (mean = 83.4), with 53% of the phylogenies including more than 50% of the described taxonomic diversity (sampling fraction mean = 57%). Clade ages range from 0.5 to 88.5 million years (Myrs) (mean = 29.9; *Figure 2—figure supplement 1*). In this dataset, amphibian phylogenies are significantly larger than those of other clades ($p<0.05$) (*Figure 2—figure supplement 1*). Amphibian and squamate phylogenies are also significantly older ($p<0$). Groups also differ in sampling fraction: plant ($p<0.01$) and squamate ($p<0$) phylogenies are significantly less sampled than phylogenies of other groups. Our dataset triples the data presented in previous meta-analyses of the Neotropics in terms of number of species, for example, 214 clades and 4450 species in *Antonelli et al., 2018c*, and quadruples it in terms of sampling, with 20.8 species per tree in *Antonelli et al., 2018c*.

## Estimating the tempo and mode of Neotropical diversification

### Diversification trends based on traditional diversification rates

To understand the tempo (Q1) and drivers of Neotropical diversification (Q2), we compared the fit of birth-death models applied to 150 phylogenies, including models where diversification rates are constant, vary through time, vary as a function of past global temperatures, or vary according to past Andean elevation (see Methods). When only models with constant diversification and time-varying rates were considered, constant models best fit 67% of the phylogenies (101 clades) (*Supplementary file 1A*). In the remaining 49 trees, we detected variation in diversification rates. Speciation decreased towards the present in 28 trees (57%), increased in 12, and remained constant (being extinction time-variable) in 9, although the proportions varied between lineages (*Figure 3a*). The proportion of clades that evolved at constant diversification decreased to 50.6% (76 clades) when the comparison included more complex environmental models (*Figure 4*; *Supplementary file 1B*; *Meseguer, 2021*). The proportion of time-variable models also increased to 74 trees.

The empirical support for the main species richness dynamics from the 150 phylogenies was as follows: gradual expansions (Sc. 1, constant diversification) were detected in 101–76 phylogenies if environmental models were considered; exponential expansions (Sc. 2, increases in diversification) were detected in 20–30 clades; and saturated expansions and declining dynamics (Sc. 3 and 4, diversification decreases) were supported in 24–31 and 5–9 clades, respectively (*Table 1* and *Figure 4*). Diversification trends remained similar when small (<20 species) or poorly sampled (<20% of the species sampled) phylogenies were excluded from the analyses (99 and 137 trees remaining, respectively), although the proportion of constant diversification models decreased in all cases (55–35%; *Figure 3—figure supplement 1*; *Figure 4—figure supplement 1*).

Rate variation was inferred from models that can capture the dependency of speciation and/or extinction rates over time (time-dependent models) or over an environmental variable (either temperature- or uplift-dependent models). Among them, temperature-dependent models explained diversification in 40 phylogenies (26.7%). Time-dependent models best fit 17 clades (11%). Uplift-dependent models explained another 11% (*Figure 5*, *Supplementary file 1B*). The relative support for time-, temperature- and uplift-dependent models remained similar regardless of whether we compared the fit of the best or second-best models (defined based on ΔAIC values; *Figure 5—figure supplement 1*), although overall support for constant-rate scenarios decreased in the latter.

Results also remained stable regardless of the paleotemperature curve (*Zachos et al., 2008*; *Hansen et al., 2013*; *Veizer and Prokoph, 2015*) considered for the analyses (*Figure 5—figure supplement 2*). Diversification analyses based on the different paleotemperature curves produced almost identical results, in terms of model selection, parameter estimates, and main diversification trends. Therefore, we present and discuss the results based on the curve of *Veizer and Prokoph,*

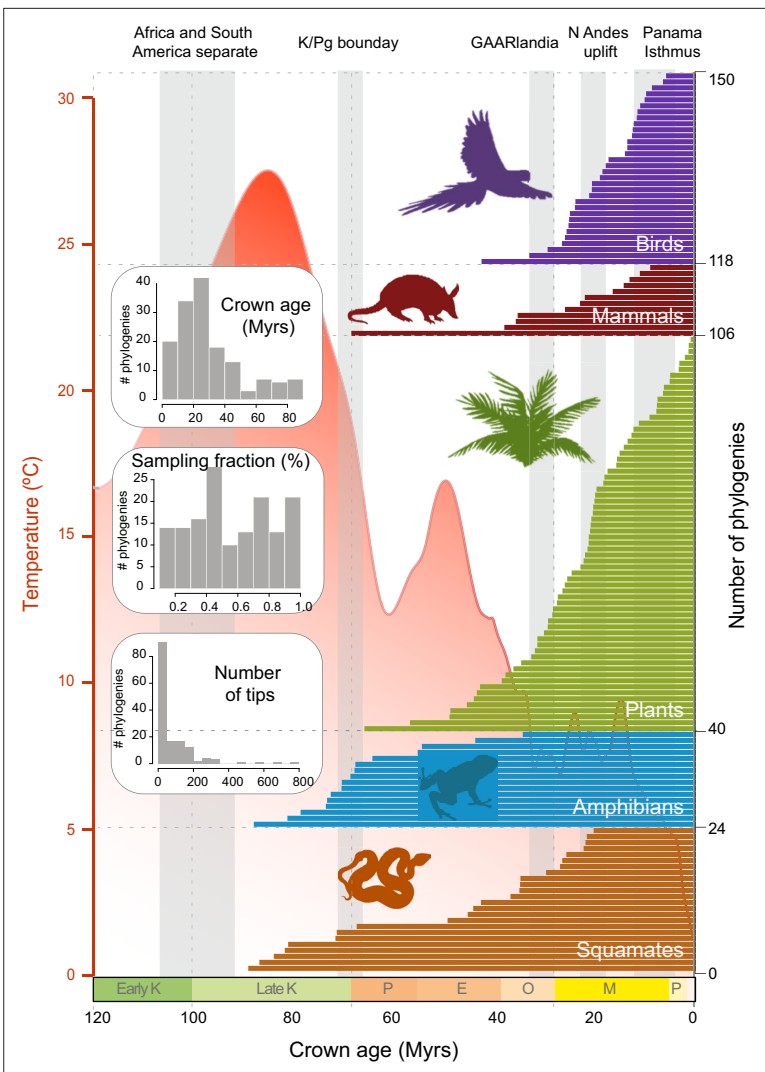

**Figure 2.** Time of origin for Neotropical tetrapods and plants. Horizontal bars represent crown ages of 150 phylogenies analysed in this study. Shaded boxes represent the approximate duration of some geological events suggested to have fostered dispersal and diversification of Neotropical organisms. Inset histograms represent summary statistics for crown age (mean = 29.9 Myrs), sampling fraction (mean = 57%), and tree size (mean = 83.4 species/tree). Mean global temperature curve from *Zachos et al., 2008*. Abbreviations: K, Cretaceous; P, Paleocene; E, Eocene; O, Oligocene; M, Miocene; P, Pliocene (Pleistocene follows but is not shown); GAARlandia, Greater Antilles and Aves Ridge. Animal and plant silhouettes from PhyloPic (http://-phylopic.org/). *Figure 2— source data 1* includes the dataset of plant, mammal, bird, squamate, and amphibian phylogenies and the original references for this data. *Figure 2—figure supplement 1* represents summary statistics for crown age, sampling fraction, and tree size for each clade. *Figure 2—figure supplement 2* includes box plots showing differences in sampling fraction, clade age, and number of species per tree for the different taxonomic groups considered in this study.

The online version of this article includes the following source data and figure supplement(s) for figure 2:

**Source data 1.** Includes the dataset of plant, mammal, bird, squamate, and amphibian phylogenies and the original references for this data.

**Figure supplement 1.** Dataset overview.

**Figure supplement 2.** Box plots showing differences in sampling fraction, clade age (i.e., crown age), and number of species per tree (i.e., tree size) for the different taxonomic groups considered in this study (Amphibia, Birds, Mammals, Plants, Squamata).

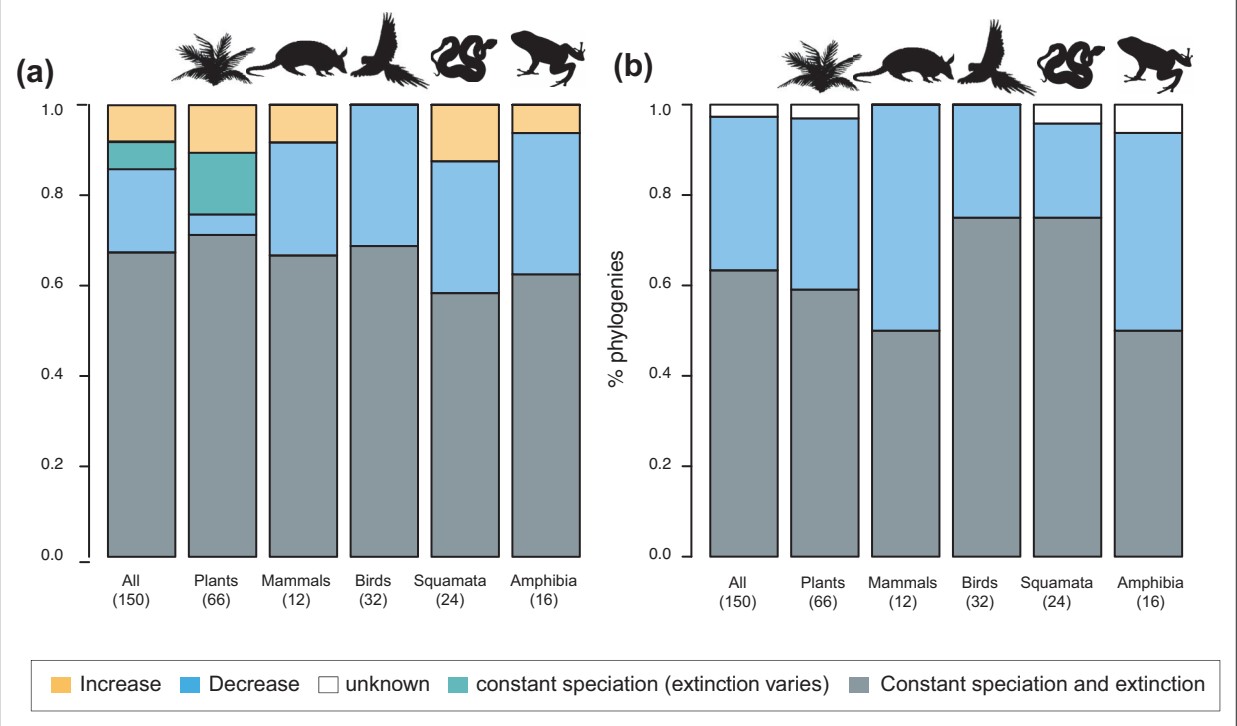

**Figure 3.** Speciation trends in 150 phylogenies of Neotropical plants and tetrapods. The histograms show the proportion of phylogenies for which constant vs. time-variable diversification models were the best fit, as derived from (**a**) canonical and (**b**) pulled diversification rates when comparing time-dependent models against constant models. In **Figure 3a**, the proportion of time-variable models is subdivided by the proportion of phylogenies in which speciation rates increase through time, decrease through time, or speciation remains constant (being extinction time-variable). In **Figure 3b**, speciation trends are derived from present-day pulled extinction rates $\mu_p(0)$: negative present-day pulled extinction rates values $(\mu_p(0)<0)$ indicate decreasing speciation trends through time (**Louca and Pennell, 2020**). Positive $\mu_p(0)>0$ values are possible under both increasing and decreasing speciation rates, in which case speciation trends are designed as 'unknown'. **Figure 3—source data 1** provides the data to construct **a** and **Figure 4a**. **Figure 3—source data 2** provides the data to construct **Figure 3b**. **Figure 3—figure supplement 1** shows the proportion of phylogenies fitting different pulled diversification models for a reduced dataset including only trees with more than 20 species (*N*=99), or with a sampling fraction over 20% (*N*=137).

The online version of this article includes the following source data and figure supplement(s) for figure 3:

**Source data 1.** Provides the data to construct **Figure 3a** and **Figure 4a**.

**Source data 2.** Provides the data to construct **Figure 3b**.

**Figure supplement 1.** Speciation trends on 150 phylogenies of Neotropical plants and tetrapods.

*2015*, as this is the only curve spanning the full time range of all the Neotropical lineages included in our dataset (150 phylogenies).

### Diversification trends based on pulled diversification rates

To gain further insights in Neotropical diversification (Q1), we explored congruent diversification models defined in terms of pulled diversification rates (PDR, $r_p$) (**Louca and Pennell, 2020**; **Louca et al., 2018**). These analyses recovered consistent diversification trends with those found above: 63% of the phylogenies (95 clades) better fit constant pulled models (**Figure 3b**; **Supplementary file 1C**). Meanwhile in 37% of the phylogenies (55 clades) we found variation in PDR through time. Diversification trends remained similar when small (<20 species) or poorly sampled (<20% of the species sampled) phylogenies were excluded from the analyses (**Figure 3—figure supplement 1**). We also detected negative pulled present-day extinction rates $\mu_p(0)$ in most of the phylogenies (51 clades, 92%) in which PDR varied through time, suggesting that speciation was decreasing. Based on PDR, we could only detect constant diversification (Sc. 1) or decreases in speciation, and thus the combined support for Sc. 2, 3, and 4 (**Table 1**).

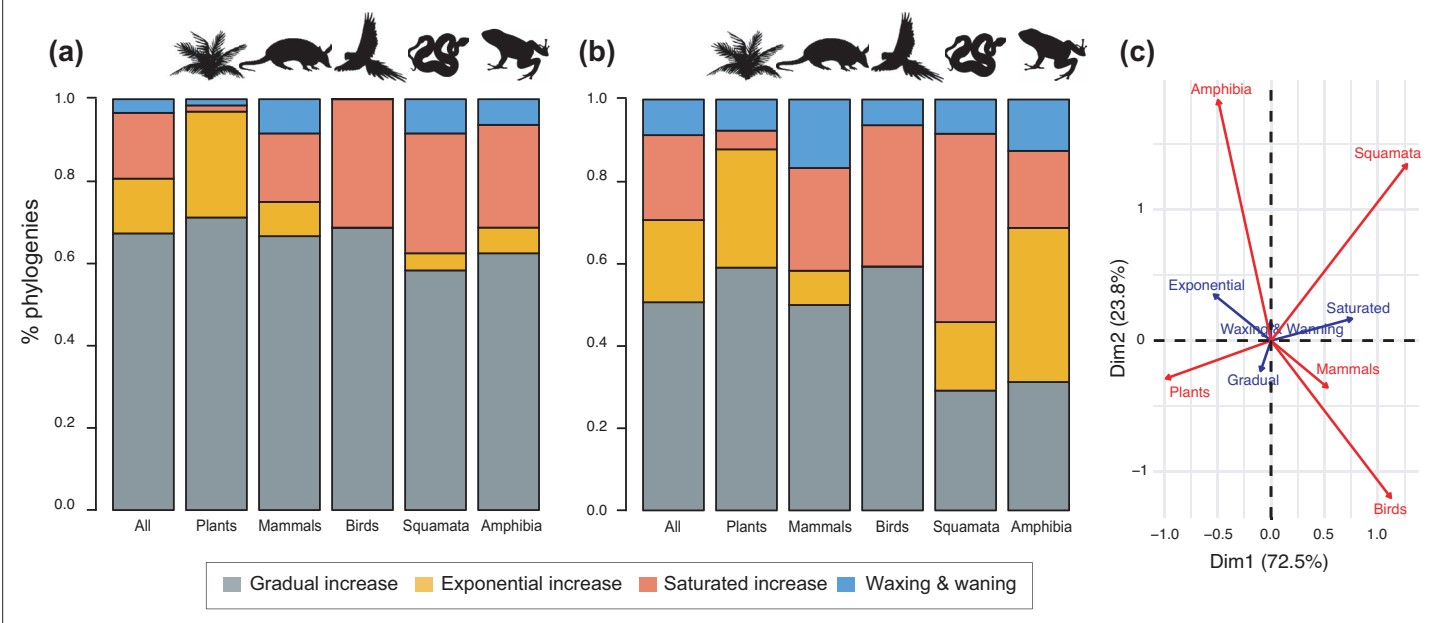

**Figure 4.** Diversity dynamics in 150 phylogenies of Neotropical plants and tetrapods. The histograms show the proportion of phylogenies for which gradual increase (Sc. 1), exponential increase (Sc. 2), saturated increase (Sc. 3), and waxing and waning (Sc. 4) scenarios were the best fit, as derived from net diversification trends when comparing (**a**) time-dependent models against constant models and (**b**) environmental (temperature- and uplift-dependent models) against time-dependent and constant models. (**c**) Correspondence analysis showing the association between species richness dynamics (represented by blue arrows) and major taxonomic groups (red arrows). If the angle between two arrows is acute, then there is a strong association between the corresponding variables. *Figure 4—source data 1* provides the data to construct *Figure 4b and c*. Source data to generate *Figure 4a* is provided as *Figure 3—source data 1*, file 2; *Figure 4—figure supplement 1* shows the proportion of phylogenies best fitting different species richness dynamics for a reduced dataset including only trees with more than 20 species (*N*=99), or with a sampling fraction over 20% (*N*=137).

The online version of this article includes the following source data and figure supplement(s) for figure 4:

**Source data 1.** Provides the data to construct *Figure 4b and c*.

**Figure supplement 1.** Species richness dynamics on 150 phylogenies of Neotropical plants and tetrapods.

## Neotropical bioregionalization

To examine the spatial variation of diversification dynamics within the Neotropics (Q3), we first had to identify geographic units of long-term Neotropical evolution suitable for comparison. We found that most clades in our study were distributed in most Neotropical WWF ecoregions (*Figure 6—source data 1*), suggesting that species presence-absence data might be of limited use for delimiting geographic units at the macroevolutionary scale of this study. In contrast, based on clades' abundance patterns, we identified five clusters of regional assemblages that represent long-term clade endemism (*Figure 6*; *Figure 6—figure supplement 1*; *Figure 6—source data 2*): cluster 1 (including the Amazonia, Central Andes, Chocó, Guiana Shield, Mesoamerica, and Northern Andes), cluster 2 (Atlantic Forest, Caatinga, Cerrado, Chaco, and temperate South America), cluster 3 (Caribbean), cluster 4 ('elsewhere' region), and cluster 5 (Galapagos). An alternative clustering (*Figure 6—figure supplement 2*) separating Mesoamerica from cluster 1, and the Chaco and temperate South America from cluster 2, received lower support (*Figure 6—figure supplement 1*).

## Variation of diversification dynamics across taxa, environmental drivers, and biogeographic units

We evaluated the prevalence of macroevolutionary scenarios 1–4 (*Figure 1*) across environmental drivers (Q2), biogeographic units (Q3) and taxonomies (Q4) (see *Methods*). *Table 2* summarize all the results. We found that species richness dynamics were related to particular environmental drivers (p=0.003; Q2). Pairwise comparisons indicated that temperature-dependent models tended to best fit clades experiencing saturating (p=0.049) and declining (p=0.05) diversity dynamics. Meanwhile,

**Table 1.** Alternative species richness dynamics (Sc. 1–4) and the corresponding diversification processes (a–c) able to explain Neotropical diversity.

Species richness dynamics represent scenarios of expanding (Sc. 1–2), saturating (Sc. 3) and contracting (Sc. 4) diversity, in which speciation ($\lambda$) and/or extinction ($\mu$) remain constant or vary through time. The number of phylogenies supporting each model is provided for all lineages pooled together, and for plants and tetrapods separately. Empirical support for each evolutionary model is based on canonical diversification rates (CDR), and pulled diversification rates (PDR), by comparing the constant model against different sets of time-variable models. For CDR, we provide as well the results (in italic) based on model comparisons including constant, time-variable, and paleoenvironmental-dependent (temperature and uplift) models.

| Diversity dynamics | CDR all (plant/tetra) | PDR all (plant/ tetra) | Diversification process | Model parameters | CDR all (plant/tetra) | PDR all (plant/ tetra) |
|---|---|---|---|---|---|---|
| **Sc 1.** Gradual increase | 101 (47/54) *76 (40/37)* | 95 (39/56) | (a) Constant $\lambda$ and $\mu$ | $\lambda(t) = \lambda_0$, $\mu(t) = \mu_0$ | 101 (47/54) *77 (40/37)* | 95 (39/56) |
| | | | (b) Equivalent increase in $\lambda$ and $\mu$ | $\lambda(t) = \lambda_0 e^{\alpha t}$, $\mu(t) = \mu_0 e^{\beta t}$, $\lambda_0 = \mu_0$, $\alpha = \beta$ | 0 (0/0) *0 (0/0)* | |
| | | | (c) Both | | 0 (0/0) *0 (0/0)* | |
| **Sc 2.** Exponential increase | 20 (17/3) *30 (19/11)* | 51 (25/26) * | (a) Increasing $\lambda$, constant $\mu$ | $\lambda(t) = \lambda_0 e^{\alpha t}$, $\alpha < 0$, $\mu(t) = \mu_0$ | 9 (7/2) *9 (8/1)* | 51 (25/26) * |
| | | | (b) Constant $\lambda$, decreasing $\mu$ | $\lambda(t) = \lambda_0$, $\mu(t) = \mu_0 e^{\beta t}$, $\beta > 0$ | 10 (10/0) *13 (11/2)* | |
| | | | (c) Both | | 1 (0/1) *8 (0/8)* | |
| **Sc 3.** Saturated increase | 24 (1/23) *31 (3/28)* | | (a) Decreasing $\lambda$, constant $\mu$ | $\lambda(t) = \lambda_0 e^{\alpha t}$, $\alpha > 0$, $\mu(t) = \mu_0$ | 24 (1/23) *29 (3/27)* | |
| | | | (b) Constant $\lambda$, increasing $\mu$ | $\lambda(t) = \lambda_0$, $\mu(t) = \mu_0 e^{\beta t}$, $\beta < 0$ | 0 (0/0) *0 (0/0)* | |
| | | | (c) Both | | 0 (0/0) *1 (0/1)* | |
| **Sc 4.** Waxing and waning | 5 (1/4) *13 (5/8)* | | (a) Decreasing $\lambda$, constant $\mu$ | $\lambda(t) = \lambda_0 e^{\alpha t}$, $\alpha > 0$, $\mu(t) = \mu_0$ | 1 (1/0) *(1/5)* | |
| | | | (b) Constant $\lambda$, increasing $\mu$ | $\lambda(t) = \lambda_0$, $\mu(t) = \mu_0 e^{\beta t}$, $\beta < 0$ | 1 (0/0) *(1/1)* | |
| | | | (c) Both | | 3 (0/4) *8 (3/2)* | |

*Pulled extinction rates ($\mu_p$) can be useful for inferring speciation trends, for example, a negative present-day pulled extinction rate ($\mu_{p(0)} < 0$) is indicative that $\lambda$ decreases through time. But the opposite is not necessarily true, that is, a positive present-day pulled extinction rate ($\mu_{p(0)} > 0$) does not necessarily indicate that $\lambda$ increases through time (**Louca and Pennell, 2020**). Based on pulled extinction, we cannot infer either if diversification dropped below 0, and thus differentiate between the two scenarios in which $\lambda$ decreases through time (3. damped increase and 4. waxing and waning dynamics). Similarly, based on pulled diversification rates, we cannot identify increases in speciation or time changes in extinction rates (scenarios 1b,c; 2a,b,c; 3b,c; 4b,c).

uplift- and time-dependent models tended to best fit clades with exponentially increasing diversity (p=0.03) (**Figure 5c**).

In contrast, there is no evidence to suggest that species richness dynamics are related to a given geographic location when considering the whole dataset (**Figure 6c–f, Figure 7**; Q3). Results of

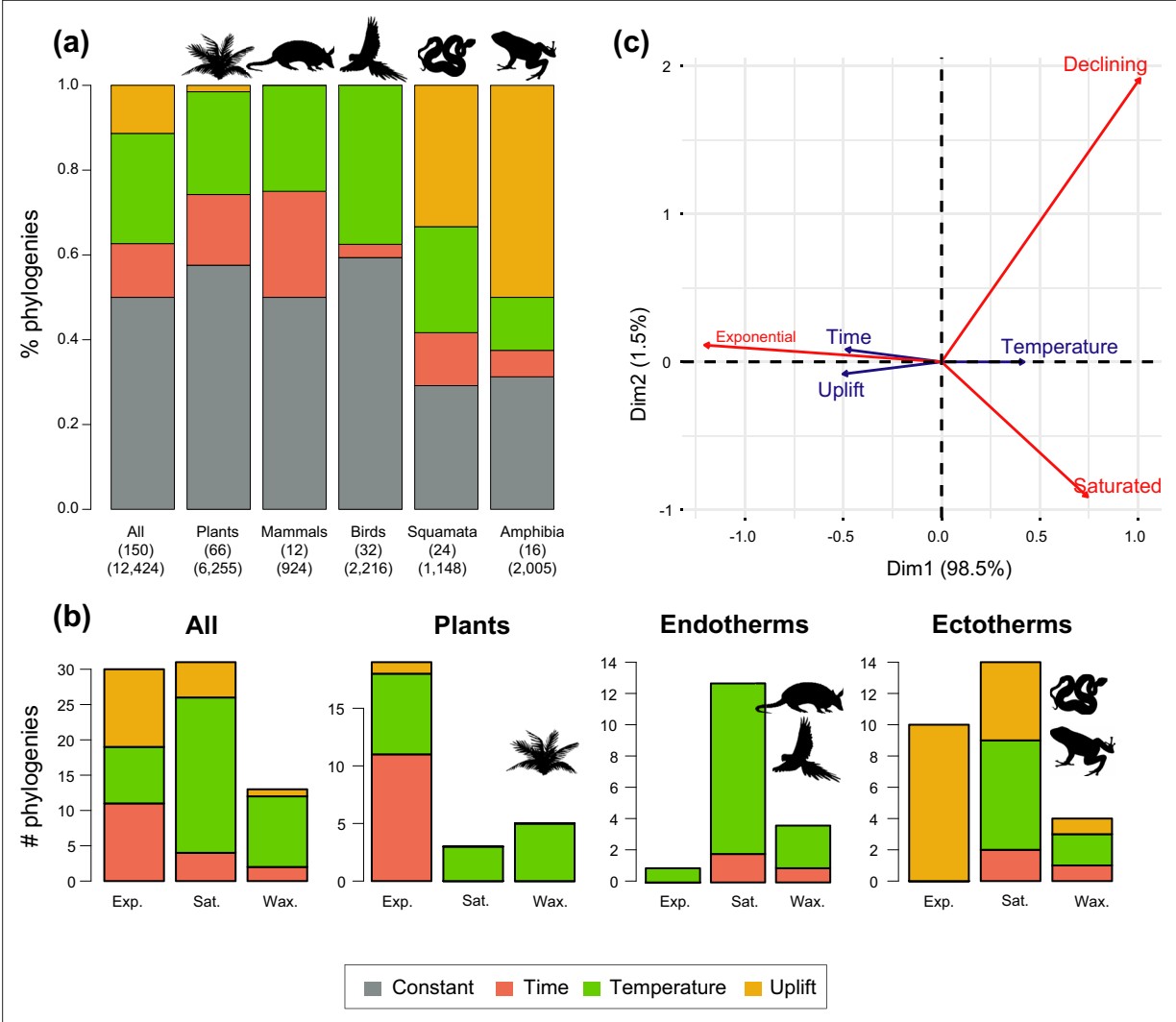

**Figure 5.** Drivers of Neotropical diversification in 150 phylogenies of Neotropical plants and tetrapods. The histograms report the proportion of (**a**) phylogenies whose diversification rates are best explained by a model with constant, time-dependent, temperature-dependent, or uplift-dependent diversification. The number of phylogenies (and species) per group is shown in parentheses. (**b**) The histograms report the number of phylogenies whose diversification rates are best explained by a model with constant, time-, temperature-, or uplift-dependent diversification according to different species richness scenarios (Exp = Exponential increase [Sc.2], Sat = Saturated increase [Sc.3], and Wax = Waxing and waning [Sc.4]), for plants, endotherm tetrapods, ectotherms, and all clades pooled together. (**c**) Correspondence analysis for the pooled dataset showing the association between species richness dynamics (represented by red arrows) and the environmental drivers (blue arrows). If the angle between two arrows is acute, then there is a strong association between the corresponding variables. *Figure 5—source data 1* provides the data to construct this figure. *Figure 5—figure supplement 1* shows the proportion of phylogenies best fitting different paleoenvironmental models based on the most supported and second most supported model. Results are also reported for a reduced dataset including only trees with more than 20 species (*N*=99), or with a sampling fraction over 20% (*N*=137). *Figure 5—figure supplement 2* shows the comparison of diversification results based on different paleotemerature curves.

The online version of this article includes the following source data and figure supplement(s) for figure 5:

**Source data 1.** Provides the data to construct this figure.

**Source data 2.** Shows diversification results for the most supported (lowest AIC value), and the second most supported diversification model.

**Figure supplement 1.** Drivers of Neotropical diversification.

**Figure supplement 2.** Result comparisons when the temperature dependency of diversification rates is estimated based on the paleotemperature curve of *Veizer and Prokoph, 2015*, or *Cramer et al., 2011* or *Hansen et al., 2013*, see Methods for details.

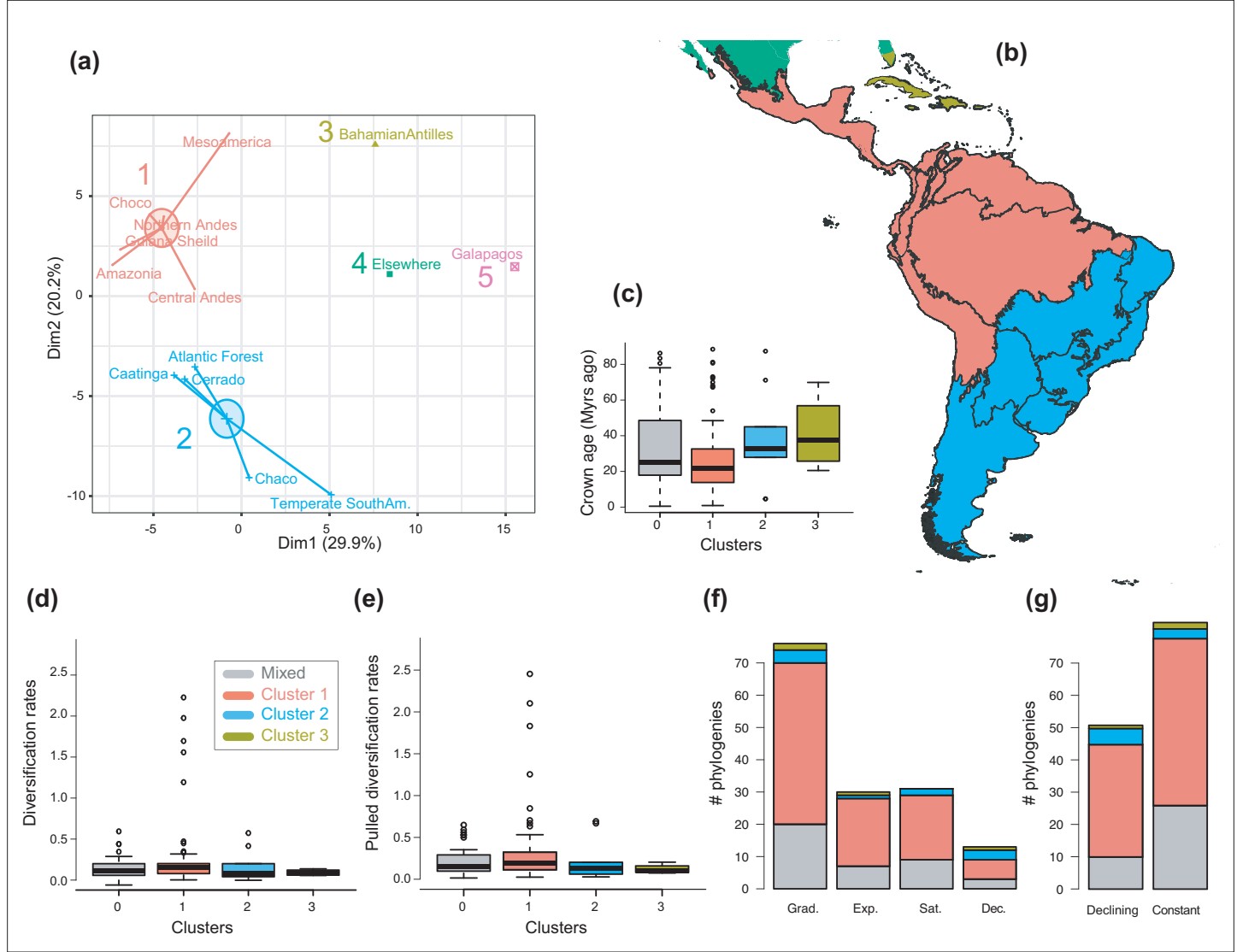

**Figure 6.** The geographical structure of long-term Neotropical diversification. (**a**) Principal component analysis (PCA) representation of the five biogeographic clusters identified based on K-means clustering of 13 areas (WWF ecoregions) and 150 clades. (**b**) Resulting clusters (1–5) in geographic space. Colours correspond with the biogeographic clusters in (**a**). Thick lines delineate the original 13 ecoregions used in the analyses. (**c**) Box plot showing differences in crown age of the phylogenies distributed in each of the biogeographic clusters. (**d**) Variation in diversification and (**e**) pulled diversification rates (derived from the constant-rate model) across geographic clusters. (**f**) Number of phylogenies for which species richness scenarios Sc. 1–4 (Grad = Gradual increase [Sc.1], Exp = Exponential increase [Sc.2], Sat = Saturated increase [Sc.3], and Dec = Declining diversity [Sc.4]) were the best fit, across geographic clusters as derived from canonical diversification rates. (**g**) Number of phylogenies for which constant vs. declining speciation rates were the best fit, across geographic clusters as derived from pulled diversification rates. *Figure 6—source data 1* provides the original data to conduct K-means clustering analyses, and generate *Figure 6a*; *Figure 6—source data 2* provides the assignation of clades to biogeographic clusters; *Figure 6—source data 3* provides the data to generate (**c, d, f**), and *Figures 7–9*; *Figure 6—source data 4* provides the data to generate *Figure 6*; for example, *Figure 6—figure supplement 1* shows the Elbow curve for K-means clustering results; *Figure 6—figure supplement 2* shows biogeographic clustering and diversification results if seven clusters are considered.

The online version of this article includes the following source data and figure supplement(s) for figure 6:

**Source data 1.** Provides the original data to conduct K-means clustering analyses, and generate *Figure 6a*.

**Source data 2.** Provides the assignation of clades to biogeographic clusters.

**Source data 3.** Provides the data to generate *Figure 6c, d, f*, and *Figures 7–9*.

**Source data 4.** Provides the data to generate *Figure 6*; for example.

**Figure supplement 1.** Elbow curve for K-means clustering results.

**Figure supplement 2.** The geographic structure of long-term Neotropical diversification.

**Table 2.** Summary p value results derived from the analysis of canonical diversification (*r*) and pulled diversification (*r*ₚ) rates.

Significant differences in the proportion of clades experiencing different diversity trajectories (based on canonical diversification rates: gradual expansions, exponential expansions, saturation or declining diversity; based on pulled diversification rates: expanding vs. declining speciation) across biogeographic units, elevations, taxonomic groups, and environmental drivers as derived from Fisher's exact tests. Significant differences in net diversification, pulled diversification, and speciation rates across biogeographic units, elevations and taxonomic groups derive from *Kruskal-Wallis chi-squared* analyses. Significant results are highlighted in bold.

| | Diversity trajectories | | Diversification rates | | Speciation rates |
|---|---|---|---|---|---|
| | *r* | *r*ₚ | *r* | *r*ₚ | *r* |
| Biogeographic units (5 clusters) | 0.459 | 0.252 | 0.168 | 0.083 | 0.248 |
| Biogeographic units (7 clusters) | 0.503 | 0.947 | 0.198 | 0.424 | 0.277 |
| Elevation | 0.504 | 0.839 | 0.672 | 0.277 | **0.034** |
| Elevation (lowland-montane combined) | 0.062 | 0.062 | 0.332 | 0.869 | **0.031** |
| Taxonomic groups | **0.000** | 0.126 | **0.000** | **0.000** | **0.000** |
| Environmental drivers | **0.003** | – | – | – | – |

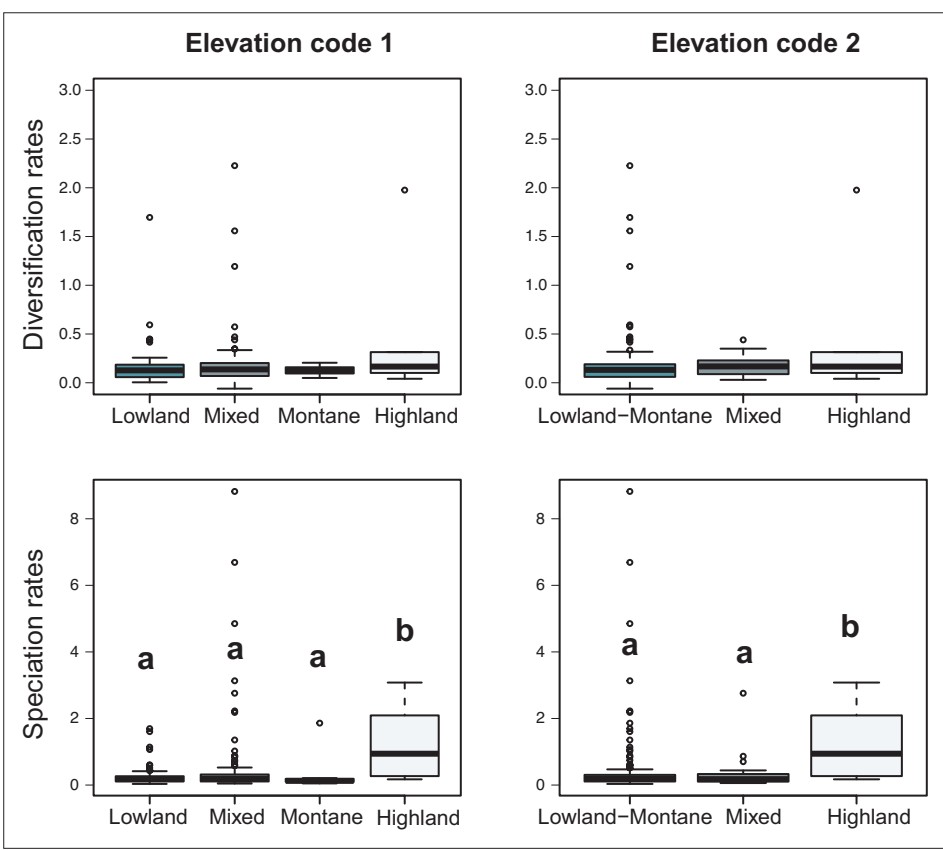

**Figure 7.** Variation in diversification rates on 150 Neotropical phylogenies of plants and tetrapods across elevation ranges. Diversification and speciation rates are derived from the constant-rate model (*Supplementary file 1A*). In the elevation code 1 the montane category has been analysed separately, while in the elevation code 2 lowland and montane categories have been pooled together (see text). Letters are used to denote statistically differences between groups, with groups showing significant differences in mean values denoted with different letters. Source data to generate this figure is provided as *Figure 6—source data 3*.

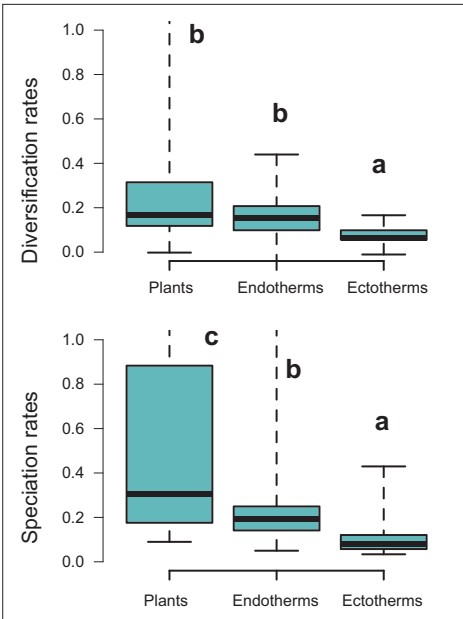

**Figure 8.** Diversification rates compared across plants and tetrapods (endotherms and ectotherms). Diversification and speciation rates are derived from the constant-rate model . Letters are used to denote statistically differences between groups, with groups showing significant differences in mean values denoted with different letters. The y-axis was cut off at 1.0 to increase the visibility of the differences between groups. Upper values for plants are therefore not shown, but the quartiles and median are not affected. Units are in events per million years. Source data to generate this figure is provided as *Figure 6—source data 3*.

Fisher's exact test show no significant differences in the proportion of clades experiencing gradual expansions, exponential expansions, saturation, or declining diversity across biogeographic units (p=0.45) or elevation ranges (p=0.062). We obtained similar results when the montane category was analysed separately (p=0.5, *Figure 7*). Diversity trajectories derived from the analysis of PDR produce the same results, with no differences in the proportion of clades experiencing constant (i.e., expanding diversity dynamics) or declining speciation trends across biogeographic units (p=0.25), or elevation ranges (p=0.062), even when the montane category was analysed separately (p=0.839). Estimates of net diversification rates (rather than diversity trajectories) derived from the constant diversification model did not differ across biogeographic units ($\chi^2$=5.05, p=0.17) or altitudinal ranges ($\chi^2$=2.20; p=0.332) either. Speciation rates did not differ between biogeographic units ($\chi^2$=4.1, p=0.25), but did vary across altitudinal ranges ($\chi^2$=6.9, p=0.03). Speciation rates were significantly higher across highland taxa (*Figure 7*). In addition, PDR did not differ across biogeographic units ($\chi^2$=6.7; p=0.083) or elevations ($\chi^2$=0.28; p=0.87).

Finally, diversity trajectories (Sc. 1–4) differed across taxonomic groups (p<0.0001, Fisher's exact test; Q4). Pairwise comparisons indicated that plants differed significantly from birds in the proportion of gradual (p<0.02), exponential (p<0.02), and saturated (p<0.0001) increase models after correcting for multiple comparisons. Birds also differed from amphibians in the proportion of saturated and exponential increases

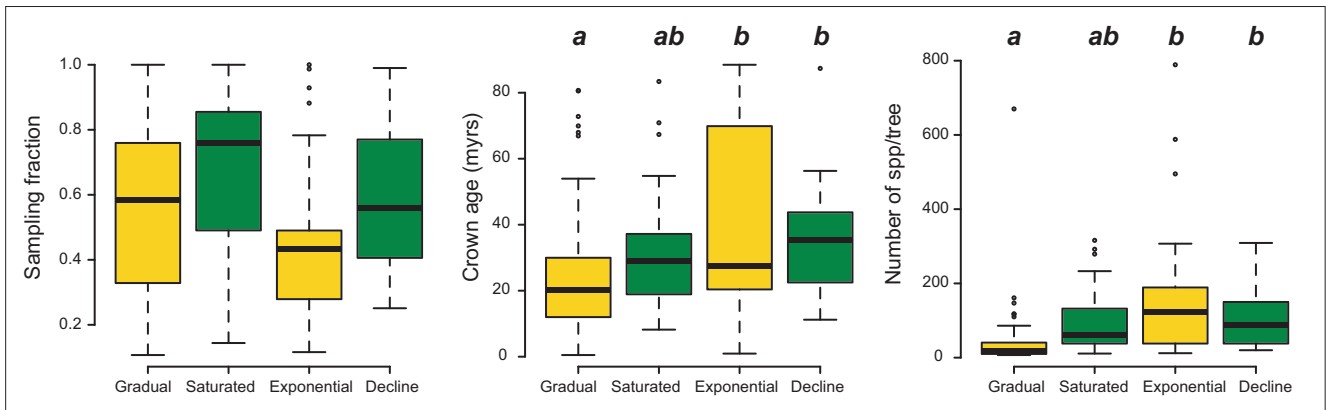

**Figure 9.** Box plots showing differences in sampling fraction, clade age (i.e., crown age), and number of species per tree (i.e., tree size) for the phylogenies supporting gradual increase (Sc. 1) vs. exponential increase (Sc. 2) vs. saturated (Sc. 3) vs. declining diversity dynamics (Sc. 4). Sampling fraction does not differ significantly between model categories, suggesting that there is no particular bias of sampling in our study. Meanwhile, there are differences in the tree size and crown age between model categories. Letters are used to denote statistically differences between groups, with groups showing significant differences in mean values denoted with different letters. Source data to generate this figure is provided as *Figure 6—source data 3*.

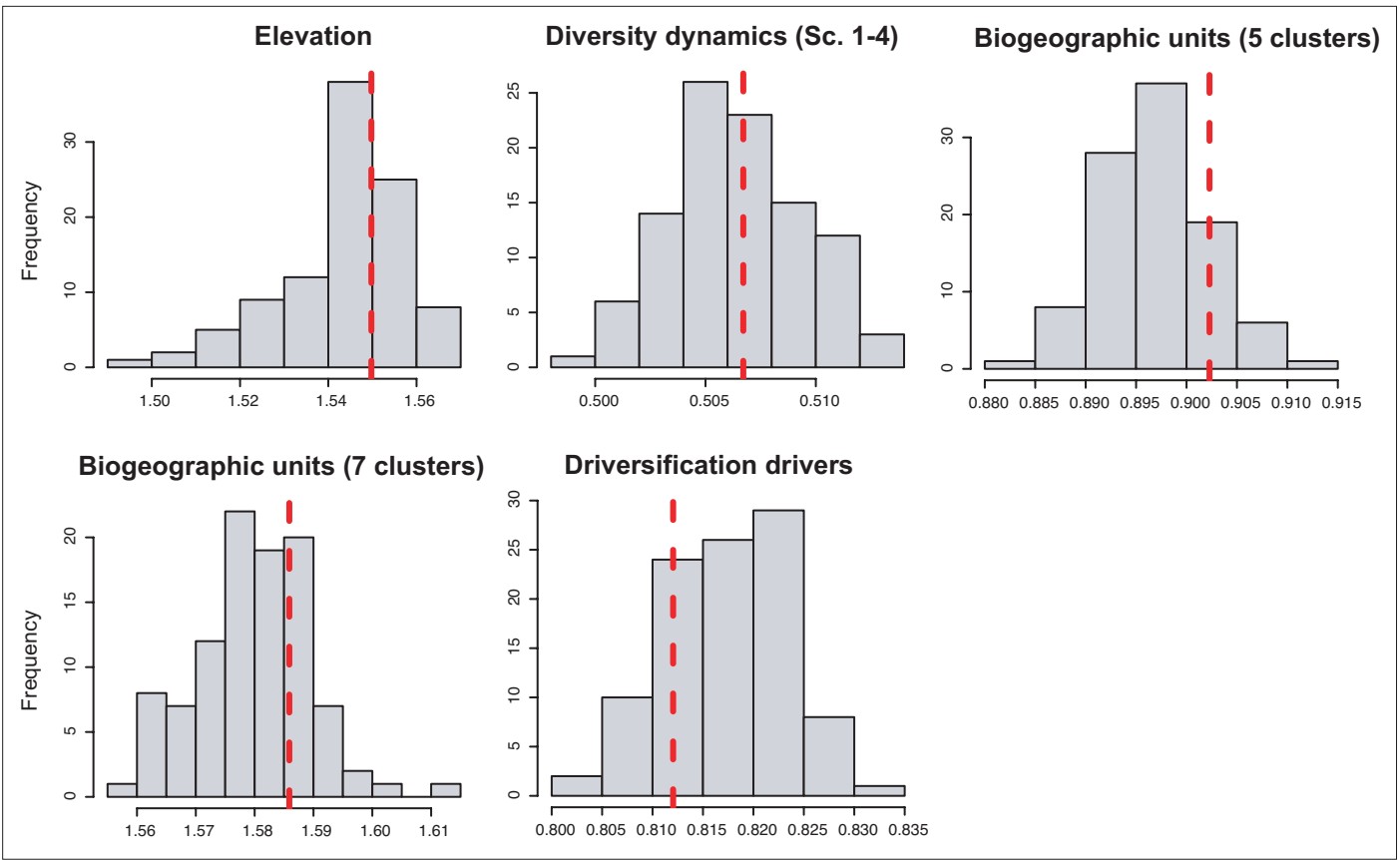

**Figure 10.** Phylogenetic signal of different multi-categorical traits. Inferred $\delta$-values (in red) compared to the distribution of values when the trait is randomized along the phylogeny.

(p<0.02). Plants differed from squamates in the proportion of exponential (p<0.0006) and saturated (p<0.008) increases (*Figure 4c*). Net diversification rates were also significantly lower for Neotropical ectotherm tetrapods than for endotherms and plants (Kruskal-Wallis chi-squared: $\chi^2$=36.7, p<0.0001) (*Figure 8*). We also found statistically significant differences in speciation rates across groups ($\chi^2$=60.8, p<0.0001): plants showed higher speciation rates than endotherms, the latter, in turn, with higher speciation rates than ectotherms.

The number of species per phylogeny differed between model categories (phylogenetic ANOVA: $F$=10.9, p=0.002). Clades fitting gradual expansion models tended to have less species than clades fitting exponential (p=0.006) and declining (p=0.03) dynamics (*Figure 9*). Taxon sampling, however, did not differ significantly (F=4.5, p=0.53). Crown age differed between model categories, being on average younger for gradual scenarios than for exponential (p=0.03) and declining (p=0.03) dynamics.

Finally, we found that no continuous ($K_r$ = 0.06, p=0.6; $K_\lambda$=0.07, p=0.4; $K_{rp}$ = 0.07, p=0.6) or multi-categorical trait displays phylogenetic signal (*Figure 10*), suggesting that the distribution of trait values is not explained by the phylogeny itself.

## Discussion

### Diversification dynamics in the Neotropics

Neotropical biodiversity has long been considered as being in expansion through time due to high rates of speciation and/or low rates of extinction (*Stebbins, 1974*; *Harvey et al., 2020*; *Meseguer et al., 2020*). Yet, to our knowledge, the generality of this trend in the Neotropics has not yet been evaluated or quantified. The higher support for the expanding diversity trend found here aligns with these ideas because most Neotropical clades (between 80% and 70%, if environmental models are considered) displayed expanding diversity dynamics through time (*Figure 4*; *Table 1*). Most of these

clades experienced a gradual accumulation of lineages (Sc. 1; between 67% and 50%), and a lower proportion (14% and 16%) expanded exponentially (Sc. 2), thus diversity accumulation accelerated recently. Results based on PDR support these conclusions, with the largest proportion of clades expanding diversity (63%) due to gradual increases (Sc. 1; *Figure 3*).

Our results, however, also provide evidence that cradle/museum models are not sufficient to explain Neotropical diversity. Based on traditional diversification rates, 16–21% of the Neotropical clades, mostly tetrapods, underwent a decay in diversification, hence a slower accumulation of diversity towards the present (Sc. 3). While a pervasive pattern of slowdowns in speciation has been described at various geographic and taxonomic scales, for example, *Morlon et al., 2010*; *Phillimore and Price, 2008*; *McPeek, 2008*; *Luzuriaga-Aveiga and Weir, 2019*, Neotropical tetrapod diversity levels have only rarely been perceived as saturated (*Santos et al., 2009*; *Harvey et al., 2020*; *Phillimore and Price, 2008*; *Weir, 2006*). Furthermore, waxing-and-waning dynamics (Sc. 4) also characterize the evolution of 3–9% of the Neotropical diversity, consistent with paleontological studies (*Hoorn et al., 1995*; *Jaramillo et al., 2006*; *Antoine et al., 2017*). We found that the species richness of five plant and eight tetrapod clades declined towards the present (e.g., *Sideroxylon* [Sapotaceae], *Guatteria* [Annonaceae], caviomorph rodents, Thraupidae birds, or Lophyohylinae [Hylidae] frogs). This proportion might seem minor but is noteworthy when compared with the low support for this model found in the Neotropical literature, which could be explained by the difficulties in inferring negative diversification rates based on molecular phylogenies (*Rabosky, 2010*). Inferring diversity declines is challenging, and often requires accounting for among-clade rate heterogeneity (*Morlon et al., 2011*). As shown here, incorporating environmental evidence could also help identify this pattern, increasing support for this scenario relative to the comparisons without these models (*Figure 4*).

Clade age and size can partially explain the better fit of the constant diversification model, thus the gradually expanding trend (Sc. 1). However, these tree features cannot explain the relative support between time-varying increasing (Sc. 2) versus decreasing (Sc. 3, 4) scenarios (*Figure 9*). Constant diversification prevails among recently originated and species-poor clades in our study, as also shown in *Condamine et al., 2019b*, which could suggest that these clades had less time to experience changes in diversification. Alternatively, the power of birth-death models to detect rate variation decreases with the number of species in a phylogeny, as shown with different diversification approaches (*Burin et al., 2019*; *Davis et al., 2013*; *Lewitus and Morlon, 2018*), suggesting that tree size could hinder the finding of rate-variable patterns. The main patterns found in this study appear to be robust to sampling artefacts. The support for the expanding diversity scenario persisted (72–60% of clades) after excluding small trees from the analyses (<20 species; *Figure 4—figure supplement 1*). Then, the relative support for the exponentially expanding scenario (Sc. 2) increased at the expense of the gradually expanding scenario (Sc. 1), strengthening the generality of the expanding trend in the Neotropics.

Incomplete taxon sampling may flatten out lineages-through-time plots towards the present and artificially increase the detection of diversification slowdowns (*Cusimano and Renner, 2010*). If this artefact affected our results, we would expect to see that under-sampled phylogenies would tend to better fit saturated diversity models (Sc. 3). Instead, we found that sampling fraction did not differ between lineages fitting saturated versus expanding diversity models (*Figure 9*). Moreover, the proportion of clades fitting saturated models even increased (17–22%) after excluding poorly sampled phylogenies (<20% of the species sampled; *Figure 4—figure supplement 1*).

Support for decreasing diversification through time was larger when PDR were considered: 34% of the clades showed slowdowns in speciation (*Figure 3*). Based on PDR, however, we cannot infer if decay of speciation were accompanied by constant, declining or increasing extinction (*Louca and Pennell, 2020*), and thus determine the relative support for Sc. 2–4. If speciation slowdowns were accompanied by larger extinction decreases, it would be possible to recover expanding dynamics (Sc. 1, 2), but in most other cases, they would lead to declines in diversification (Sc. 3, 4). The limited interpretability of PDR prevents the extraction of further conclusions based on these rates (*Morlon et al., 2022*).

Still, our study illustrates the robustness of the diversification trend in the Neotropics to different modelling approaches. Despite parameter values varying substantially for some trees between the traditional and PDR methods (*Supplementary file 1*), a pattern also described in recent studies

(*Morlon et al., 2022*), our analyses support a macroevolutionary scenario of expanding diversity for most Neotropical clades (*Figure 4*).

## Taxon-specific patterns and drivers of Neotropical diversification

The variation in Neotropical diversification dynamics could be partially explained by the taxonomic affinity of the groups under study. Our study revealed contrasting evolutionary patterns for plants and tetrapods (*Figure 4*): diversity expansions (Sc. 1, 2) were more frequently detected in plants (~88%, 59 clades) than in tetrapods (~57%, 48 clades). In contrast, asymptotic increases (Sc. 3) were more frequent in tetrapods (33%, 28 clades) than in plants (4.5%, 3 clades; *Tynanthus* [Bignoniaceae], Chamaedoreae [Arecaceae], and Protieae [Burseraceae]). Net diversification rates were also higher in plants (*Figure 8*), in agreement with previous studies (*Hernández-Hernández et al., 2021*).

The study of PDR did not help to confirm or reject these conclusions. Rates from PDR models are significantly different between plants and animals (p≈0.00), in agreement with results based on traditional models (*Table 2*). Diversification trajectories derived from these rates are not different, with plants and animals exhibiting an equivalent fraction of phylogenies showing a decrease of speciation (*Figure 3b*). Since extinction dynamics cannot be derived from PDR models, we do not know if speciation slowdowns detected in plants were accompanied by larger extinction declines. Thus, we cannot rule out the scenario of expanding dynamics (Sc. 1, 2) for plants found based on traditional birth-death models.

Differences in the phylogenetic composition of the plant and tetrapod datasets do not explain this contrasted pattern. On average, plant phylogenies are not significantly younger or species-poorer than tetrapod phylogenies (*Figure 2—figure supplement 2*). Yet, the proportion of clades experiencing increasing dynamics is significantly higher for plants (*Figure 4*). Plant phylogenies are significantly less sampled than are tetrapod phylogenies, though, as explained above, incomplete taxon sampling tend to have the opposite effect over diversity curves: flattening out lineages-through-time plots towards the present, increasing the probability to detect saturated dynamics (*Cusimano and Renner, 2010*).

Alternatively, this contrasting evolutionary pattern may result from differential responses of plants and tetrapods to environmental changes (*Figure 5*). Global temperature change during the Cenozoic is found to be the main driver behind diversification slowdowns (Sc. 3) and declines (Sc. 4) of tetrapods, especially for endotherms (*Figure 5*). The positive correlation between diversification and past temperature in our temperature-dependent models indicates these groups diversified more during warm periods, such as the Eocene or the middle Miocene, and diversification decreased during cool periods. This result is in agreement with previous empirical studies (*Condamine et al., 2019a*; *Moen and Morlon, 2014*) and also with recent simulations showing a negative effect of climate cooling (and a positive effect of Andean orogeny; see below) on Neotropical tetrapod diversification (*Hagen et al., 2021*). According to the Metabolic Theory of Biodiversity, low temperatures can decrease enzymatic activity, generation times, and mutation rates (*Gillooly et al., 2001*), which may in turn affect diversification (*Allen et al., 2006*). Climate cooling may also decrease global productivity, resource availability, population sizes (*Mayhew et al., 2012*), or even species interactions (*Chomicki et al., 2019*). Only the New World monkeys (Platyrrhini) diversified more as temperature dropped. This could reflect the role of Quaternary events on primate speciation (*Rull, 2011*), and/or be an artefact of taxonomic over-splitting in this clade (*Springer et al., 2012*). In contrast, a few plant clades are influenced by temperature changes, with diversification increasing during the Neogene cooling (i.e., negative correlation between diversification and temperature; *Figure 5*). This opposite pattern suggests that Cenozoic environmental changes drove diversification slowdowns for some tetrapods, but stimulated plant diversification. Although Neotropical climate has been relatively stable through the Cenozoic in comparison to other regions (*Ziegler et al., 2003*; *Morley, 2007*), in the Neotropics, global cooling contributed to the expansion of several biomes, such as the alpine Paramos (*Madriñán et al., 2013*) and other open ecosystems (*Cheng et al., 2013*; *Dick and Pennington, 2019*), providing new opportunities for diversification. Higher mean speciation rates in plants than in tetrapods (*Figure 8*) could have provided plant lineages more opportunities for adaptation to changing environments (*Hughes and Eastwood, 2006*). Greater dispersal abilities in plants (*Antonelli et al., 2018c*; *Sanmartín and Ronquist, 2004*) may also explain this pattern.

Temperature changes emerge in our study as an important factor driving Neotropical diversification across macroevolutionary scales (*Antonelli and Sanmartín, 2011a*; *Condamine et al., 2019a*), but our results also reveal that this is not the only driver. A substantial proportion of diversification changes are attributed to Andean uplift and other factors (*Figure 5*). To a lesser extent, Neotropical diversification is explained by ecological limits on the number of species within a clade, which would imply that diversity is bounded by specific carrying capacities (*Rabosky, 2009*; *Etienne et al., 2012*). Among the tetrapod phylogenies supporting diversification slowdowns, time-dependent models explain 3% of them (four phylogenies; *Figure 5*, *Supplementary file 1B*), suggesting that ecological limits play a minor role in the Neotropics. Time-dependent models with decreasing speciation have been suggested to be a good approximation of diversity-dependent diversification, whereby speciation rates decline as species accumulate (*Rabosky et al., 2014*; *Morlon, 2014*). In fact, recent studies show that time- and diversity-dependent models are difficult to distinguish based on extant phylogenies (*Pannetier et al., 2021*). As discussed above, our results lend support to an alternative explanation for diversification slowdowns: the idea that tetrapods, for some periods, were less successful in keeping pace with a changing environment (*Condamine et al., 2019a*; *Moen and Morlon, 2014*).

The Andean orogeny mostly impacted tetrapod diversification (*Hagen et al., 2021*), especially ectotherms. Diversification of some lineages increased as the Andes rose, including Andean-centred lineages such as Liolaemidae lizards, but also others predominantly distributed outside the Andes, such as Leptodactylidae frogs. Sustained diversification in the context of Andean orogeny, both into and out of the Andean region, could be explained by increasing thermal and environmental gradients, from the equatorial areas to Patagonia or from west-east (*Fouquet et al., 2014*; *Moen and Wiens, 2017*). Other possible correlates include changes in elevational distributions of lineages (*Kozak and Wiens, 2010*; *Hutter et al., 2017*), or recurrent migrations (*Santos et al., 2009*; *Esquerré et al., 2019*).

In contrast to tetrapods, plant diversity expansions were primarily associated with temperature cooling and with time, where the latter represents a null hypothesis; the better fit of a time-dependent model, in comparison to environmental models, is generally indicative of factors not being investigated here (*Morlon, 2014*). Many of the plant lineages fitting time-dependent models represent textbook examples of ongoing radiations; for example, centropogonids (*Lagomarsino et al., 2016*), *Lupinus* (*Drummond et al., 2012*), or *Inga Kursar et al., 2009*, whose diversification has been associated with biotic drivers, such as the evolution of key adaptations or pollination syndromes. These factors are taxon-specific and were not evaluated in this study, where we focused on global phenomena. Similarly, we did not assess the role of the emergence of angiosperm-dominated rainforests in the evolution of tetrapods. Angiosperm-dominated forests were already established in the Neotropics by the Palaeocene (*Carvalho et al., 2021*), while the age of origin for most clades in our study postdates this period (*Figure 2*). In all cases, our results add support to the role of environmental and biotic factors as non-mutually exclusive drivers of macroevolutionary changes on Neotropical plants.

## Neotropical bioregionalization at macroevolutionary scales

Understanding the spatial variation of Neotropical biodiversity dynamics is key to understanding the determinants of the exceptional diversity of the Neotropics. The first step towards this is the identification of evolutionary arenas of Neotropical diversification.

Conventional bioregionalizations schemes, such as biomes (*Walter and Box, 1976*), ecoregions (*Olson et al., 2001*), or other pre-defined biogeographic units (*Antonelli et al., 2018c*; *Escalante et al., 2013*; *Morrone, 2014*), could represent evolutionary arenas of diversification suitable for comparison. These bioregions have often been shown to be useful for categorizing actual species ranges. However, they are less appropriate for examining clade endemism at the macroevolutionary scale. The temporal origin of several bioregions postdates the origin of many of our clades (*Figure 2*). For instance, the Cerrado is inferred to have originated during the late Miocene (*Simon et al., 2009*), and the Chocó during the Pliocene-Pleistocene (*Pérez-Escobar et al., 2019*). In addition, most clades in our study appear distributed in most Neotropical ecoregions and could not be assigned to a single region (*Figure 6—source data 1*). The lack of a clear geographical structure for taxa of higher rank could be explained by the fact that conventional bioregionalizations generally represent categorizations based on data on the contemporary distribution of species without explicitly considering ancestral distributions or the relationships among species (*Holt et al., 2013*; *Kreft and Jetz, 2010*).

We propose an alternative bioregionalization scheme of the Neotropical region that accounts for long-term regional assemblages at macroevolutionary scales (*Figure 6*). We identify five biogeographic units that represent macroregions where different independent Neotropical radiations occurred over millions of years of biotic evolution. These regions are defined in terms of species richness patterns within clades (*Figure 6—source data 1*; *Figure 6—source data 2*), showing that species-rich clades in Amazonia also tend to be species-rich in the Andes, Chocó, Guiana Shield, and Mesoamerica (biogeographic cluster 1), without excluding that some species within these clades occur in other regions. Meanwhile, clades that are species-rich in the Atlantic Forest tend to be rich in the Caatinga, Cerrado, Chaco, and temperate South America (cluster 2). This regionalization roughly coincides with the Neotropical sub-regions proposed in previous studies (*Morrone et al., 2022*). The biogeographic cluster 1 corresponds with a broad 'pan-Amazonian' region that relied on the ancient Amazon Craton (*Hoorn et al., 2010*). Cluster 2 broadly groups different formations of the area known as the 'Dry Diagonal' (*Prado and Gibbs, 1993*; *Luebert, 2021*), which are geologically younger, dating from the Miocene (*Pennington et al., 2006*; *Beerling and Osborne, 2006*; *Becerra, 2005*). Although lineage crown ages do not differ between these regions (*Figure 6*). Clusters 1 and 2 include regions identified as transition zones in previous studies – Mesoamerica and temperate South America, respectively (*Kreft and Jetz, 2013*). Our analyses merged these regions with the core area with which it showed the greatest affinity, although other less supported classification schemes separate transition regions into individual clusters (*Figure 6—figure supplement 1*, *Figure 6—figure supplement 2*). Within each of these clusters, the contribution of in situ diversification is therefore more relevant than dispersion to explain their biotic assemblage. As such, these biogeographical clusters form distinctive units of Neotropical evolution and represent long-term clade endemism.

## The geographical structure of Neotropical diversification

The variation in Neotropical diversification dynamics described in this study (*Figure 4*) could not be explained by geography. We did not find evidence to reject the null hypothesis of equal diversification, with similar diversity dynamics (Sc. 1–4) found across the biogeographic units of Neotropical evolution identified here (*Figure 6*, *Table 2*). We obtained the same result when Mesoamerica and temperate South America transition zones were analysed separately (*Figure 6—figure supplement 2*). In addition, we did not find differences in diversification dynamics between elevational ranges. These results were consistent whether we analysed net diversification rates or their derived diversity trends (Sc. 1–4). In the former, Neotropical lineages distributed in different elevations did differ in their speciation rates, as found in previous studies: speciation increased with altitude (*Drummond et al., 2012*; *Weir, 2006*; *Quintero and Jetz, 2018*; *Vasconcelos et al., 2020*; *Rahbek et al., 2019*). Elevated speciation rates might result from ecological opportunities on newly formed high-altitude environments, or those newly exposed after periods of cooling (*Armijo et al., 2015*; *Blisniuk et al., 2005*; *Flantua et al., 2019*). However, elevated speciation rates were also accompanied by elevated extinction in these habitats, hence net diversification remains comparable. The hypothesis of comparable diversification was also supported when comparing PDR (*Figure 6*). Geographic diversification may vary within taxonomic groups, though small sample sizes prevent us from drawing any firm conclusions on this.

The use of clades (rather than species) as evolutionary units in our biogeographic comparisons is original, and allowed to compare linage diversification trends through time (i.e., constant, expanding, declining) across regions, and not just present-day diversification rates, as in different comparable studies focused at the species level, for example, *Harvey et al., 2020*; *Quintero and Jetz, 2018*; *Smith et al., 2014*. Present-day diversification rates are structured geographically in the Neotropics (*Harvey et al., 2020*; *Quintero and Jetz, 2018*; *Rangel et al., 2018*), but our study shows that present diversification does not represent long-term evolutionary dynamics. The lack of a clear geographic structure of long-term diversification suggests that the evolutionary forces driving diversity in the Neotropics acted at a continental scale when evaluated over tens of millions of years. Evolutionary time and extinction could have eventually acted as levelling agents of diversification across the Neotropics over time.

These results also suggest that differences in species richness between the Neotropical bioregions defined here might not be attributable to long-term differences in diversification rates, nor to differences in diversification dynamics. Nor could time alone explain these differences, as we found

no significant differences in the crown age of the phylogenies distributed in the different biogeographic clusters (*Figure 6*). Several studies have highlighted the role of dispersal in the configuration of modern Neotropical biotas (*Carrillo et al., 2020*; *Smith et al., 2014*; *Bacon et al., 2015*; *Antonelli et al., 2015*). By focusing exclusively on Neotropical radiations, we did not consider the role of dispersal into and out of the Neotropics (or within Neotropical regions) as an additional factor explaining Neotropical diversification. Future studies integrating biogeographic and diversification processes will be needed to provide a complete picture on the drivers of Neotropical diversification.

## Limitations and perspectives

The results and conclusions presented here represent our best attempt to infer complex processes in deep geological times, and need to be interpreted in light of the general challenges in estimating diversification rates from phylogenies of extant species. *Louca and Pennell, 2020*, have reanimated this debate by showing that there is an infinite number of 'congruent' models that yield the same likelihood for any combinations of speciation and extinction rates. However, when speciation and extinction rates are defined as functions of time and constrained to follow specific functional forms, such as the exponential or a biologically motivated function (such as the environmental dependency tested here), speciation and extinction rates are identifiable (*Morlon et al., 2022*). The time-dependent models we applied have been shown to perform well in recovering speciation and extinction parameters, including negative net diversification (*Morlon et al., 2011*), detecting shifts of diversification (with regularization techniques as proposed in *Morlon et al., 2022*), and correctly identifying the diversification model and paleodiversity dynamic (*Mazet et al., 2022*). The same applies to environment-dependent models (*Lewitus and Morlon, 2018*).

According to previous simulations, it is unclear whether temperature-dependent models can be accurately distinguished when the effect of the environmental dependence on diversification is weak. Model selection tends to be sensitive when dependency values ranges between –0.1 and 0.1 (*Lewitus and Morlon, 2018*). In these cases, constant-rate models tend to overfit, which means that we are conservative when we conclude that temperature-dependent models are estimated as best fitting in our study. We therefore measured the impact this bias might have on our results, expecting that if the constant-rate model overfits, we would observe that the temperature-dependent model is more often ranked second in the selection procedure. Of the 76 clades with a constant-rate model as the best fit, our results indicate that 50% (38/76) have temperature-dependent models as the second best fit, 40% (30/76) have time-dependent models, and 10% (8/76) have Andean-dependent models (*Figure 5—source data 2*). This suggests that there is no clear bias against temperature-dependent models. Furthermore, when evaluating the dependency values of the 38 clades that are best fit by a constant-rate model and second best fit by a temperature-dependent model, we find that only 26% (10/38) have dependency values ranging between –0.1 and 0.1 for the temperature models. These 10 trees represent 6% of our dataset, suggesting that there is a low proportion of trees susceptible to suffer from this bias.

In our study, the relative support for time-, temperature- and uplift-dependent models remained stable to AIC variations (*Figure 5—figure supplement 1*). Model support also remained stable regardless of the paleotemperature curve considered for the analyses (*Figure 5—figure supplement 2*). Furthermore, the use of an hypothesis-driven framework has been suggested as a potential solution to alleviate the problem of non-identifiability of diversification parameters, by setting up explicit prior assumptions and delimiting the potential parameter space (*Louca and Pennell, 2020*; *Morlon et al., 2022*; *Magee et al., 2020*). Here, we do not evaluate every possible factor that could potentially explain Neotropical biodiversity, but only confront scenarios capturing well-established hypotheses on Neotropical diversification. We focus on the role of the Cenozoic change in Andean elevation and climatic oscillations because they have previously been pinpointed as essential for explaining Neotropical biodiversity (*Hoorn et al., 2010*; *Rangel et al., 2018*; *Hagen et al., 2021*). Thus, our main interest is to explore which of these factors likely explains the data compiled, although other factors could have played a role.

We have compiled as many Neotropical clades (and as many species per clade) as possible, resulting in a phylogenetic dataset representing, to our knowledge, one of the largest assembled to date. Yet, we are keenly aware that we still come up short, especially with the plant database. Our plant dataset (>6000 species, 66 clades) includes just a small fraction (~7% of the total species) of the vast diversity

described in the region. As such, our results, which show contrasting diversification dynamics between plants and tetrapods, should be taken with caution. Future investigations would be necessary to confirm the generality of the expanding trend for plants. Basic knowledge of the real Neotropical diversity (and phylogenetic relationships) also remains incomplete, for example, *Kier et al., 2005*, and we anticipate the discovery of additional patterns by expanding the database.

Similarly, we did not manage to sample evenly across all regions. Our conclusions on the spatial patterns of diversification are derived from the study of a fraction of the Neotropical diversity, where tropical rainforest lineages from the broad 'pan-Amazonian' region are most abundant. Although sample size in our biogeographic comparisons is large (150 observations), some categories of these variables are poorly represented, which might limit the performance of some statistical tests. For instance, there are 97 phylogenies assigned to the biogeographic cluster 1, while only 10 in cluster 2. Note that there are other clades (39) containing species on poorly represented regions that fall in the 'mixed' category, as they share species with different areas. Our sampling, however, includes representatives from all the main regions in the Neotropics. Yet, we did not identify a common diversification trajectory or diversification rates, among the fewer clades distributed on poorly represented regions (e.g., southern South America clades experienced all gradual, exponential, and declining dynamics, as did the clades from other regions; *Figure 6*). It is also reasonable to assume that our sampling reflects a fair proportion of species per region, considering the extension of these regions in the Neotropics and the representativeness of our dataset; at least for tetrapods, it includes ~60% of all described species.

Although these limitations are likely to bias our study, we deem the representativeness of our dataset, and the diversification models compared here, as adequate to support the general patterns and conclusions inferred in this study. We hope that our study will provide interesting and testable perspectives for future investigations in the Neotropics and other regions.

## Conclusions

This study represents a quantitative assessment of the prevailing macroevolutionary dynamics in the Neotropics, and their drivers, at continental and large temporal scales. Neotropical diversity has mostly expanded through time, but scenarios of saturated and declining diversity also account for a substantial proportion of Neotropical diversity. This variation in diversity trends is better explained by taxonomic rather than geographic factors, suggesting that the modern diversity observed in seed plants and tetrapods is partly a consequence of the contrasting diversification dynamics of these groups. Applying both traditional and pulled birth-death models to all phylogenies, we have shown a good consistency in the inferred models, which suggests that our study can provide meaningful estimates of diversification.

Whether the main pattern of diversity expansion over time can contribute to explain why the Neotropics have more species than other regions in the world remains to be evaluated based on comparative data from other regions (*Antonelli et al., 2015*; *Couvreur, 2015*). Such a comparison could reveal contrasted diversity trajectories in different continents and help to elucidate the association between current diversity levels and long-term diversity dynamics.

## Methods
### Data compilation

Neotropical clades, representing independent radiations in the Neotropics, were pulled from large-scale time-calibrated phylogenies of frogs and toads (*Hutter et al., 2017*), salamanders (*Pyron et al., 2013*; *Pyron, 2014*), lizards and snakes (*Pyron and Burbrink, 2014*), birds (*Jetz et al., 2012*) (including only species for which genetic data was available), mammals (*Bininda-Emonds et al., 2007*; *Kuhn et al., 2011*), and plants (*Zanne et al., 2014*). To identify independent Neotropical radiations, species in these large-scale phylogenies were coded as distributed in the Neotropics – delimited by the World Wide Fund for Nature WWF (*Olson et al., 2001*) – or elsewhere using the R package *speciesgeocodeR* 1.0–4 (*Töpel et al., 2017*), and their geographical ranges extracted from the Global Biodiversity Information Facility 'GBIF' (https://www.gbif.org/), the PanTHERIA database (https://omictools.com/pantheria-tool), BirdLife (http://www.birdlife.org), and eBird (http://ebird.org/content/ebird), all accessed in 2018, in a procedure similar to *Meseguer et al., 2020*. Next, we pruned the trees

to extract the most inclusive clades that contained at least 80% Neotropical species, as previously defined. This procedure ensures that the diversification signal pertains to the Neotropics. In addition, phylogenies of particular lineages not represented in the global trees (or with improved taxon sampling) were obtained from published studies or reconstructed de novo in this study (for caviomorph rodents, including 199 species; *Supplementary file 2*). In the case of plants and mammals, most phylogenies were obtained from individual studies, given the low taxon sampling of the plant and mammal large-scale trees. However, whenever possible, we extracted phylogenies from a single dated tree rather than performing a meta-analysis of individual trees from different sources (*Hoorn et al., 2010*; *Jansson et al., 2013*), such that divergence times would be comparable. The resulting independent Neotropical radiations could represent clades of different taxonomic ranks. We did not perform any specific selection on tree size, crown age, or sampling fraction, but tested the effect of these factors on our results.

## Estimating the tempo and mode of Neotropical diversification

### Diversification trends based on traditional diversification rates

We compared a series of birth-death diversification models estimating speciation ($\lambda$) and extinction ($\mu$) rates for each of the 150 phylogenies with the R package *RPANDA* 1.9 (*Morlon et al., 2016*) (Q1). To make these results comparable with those derived from PDR below, we followed a sequential approach by including models of increasing complexity. We first fitted a constant-rate birth-death model and compared it with a set of three models in which speciation and/or extinction vary according to time (*Morlon et al., 2011*): $\lambda(t)$ and $\mu(t)$. For time-dependent models, we measured rate variation for speciation and extinction rates with the parameters $\alpha$ and $\beta$, respectively: $\alpha$ and $\beta > 0$ reflect decreasing speciation and extinction towards the present, respectively, while $\alpha$ and $\beta < 0$ indicate the opposite, increasing speciation and extinction towards the present.

We further compared constant and time-dependent models, described above, with a set of environment-dependent diversification models that quantify the effect of environmental variables on diversification (Q2) (*Condamine et al., 2013*). Environmental models extend time-dependent models to account for potential dependencies between diversification and measured environmental variables, for example, speciation and extinction rates can vary through time and both can be influenced by environmental variables. We focus here on mean global temperatures and Andean uplift. Climate change is probably one of the most important abiotic factors affecting biodiversity, of which global fluctuation in temperatures is the main component (*Prokoph et al., 2008*). In addition, the orogenesis of the Andes caused dramatic modifications in Neotropical landscapes and has become paradigmatic for explaining Neotropical biodiversity (*Hoorn et al., 2010*).

We fitted three environmental models in which speciation and/or extinction vary continuously with temperature changes ($\lambda[T]$ and $\mu[T]$), and three others with the elevation of the Andes ($\lambda[A]$ and $\mu[A]$). In this case, $\lambda_0$ ($\mu_0$) is the expected speciation (extinction) rate under a temperature of 0°C (or a paleo-elevation of 0 m for the uplift models). We also analysed whether the speciation ($\alpha$) and extinction ($\beta$) dependency were positive or negative. For temperature models, $\alpha(\beta) > 0$ reflects increasing speciation (extinction) with increasing temperatures, and conversely. For the uplift models, $\alpha(\beta) > 0$ reflects increasing speciation (extinction) with increasing Andean elevations, and conversely. We accounted for missing species for each clade in the form of sampling fraction ($\rho$) (*Morlon et al., 2011*) and assessed the strength of support of the models by computing Akaike information criterion (AICc), $\Delta$AICc, and Akaike weights (AIC$\omega$) to select the best fit model. We derived diversity dynamics (Sc. 1–4) based on the inferred diversification trends according to *Figure 1*.

For Andean paleo-elevations we retrieved a generalized model of the palaeo-elevation history of the tropical Andes, compiled from several studies (*Lagomarsino et al., 2016* and references therein). The elevation of the Andes could have indirectly impacted the diversification of non-Andean groups. We thus applied uplift models to all clades in our study. Temperature variations during the Cenozoic were obtained from (i) global compilations of deep-sea oxygen benthic foraminifera (bf) isotope ratios ($\delta^{18}O_{bf}$) (*Zachos et al., 2008*; *Prokoph et al., 2008*). This curve estimated by *Prokoph et al., 2008*, *Veizer and Prokoph, 2015*, and *Zachos et al., 2008*; *Zachos et al., 2001* provides estimates for the last 540 Myrs, thus spanning the full time range over which Neotropical lineages diversified. However, recent investigations derived other paleotemperature curves for the Cenozoic (*Hansen et al., 2013*; *Veizer and Prokoph, 2015*; *Cramer et al., 2011*). To account for the uncertainty on

global paleotemperatures on our results, we performed additional diversification analyses using other two different global curves; (ii) the temperature curve by *Cramer et al., 2011*; *Cramer et al., 2009*, which is similar to the more widely used previous curve but accounts for fluctuations in sea water (sw) $\delta^{18}O_{sw}$ through time and correct for ice volume. This curve provides temperature estimates for the last 62.4 Myrs; and (iii) the paleotemperature curve estimated by *Hansen et al., 2013*, for the last 65.6 Myrs, which accounts for ice volume and deep ocean temperature changes, and provides estimates of surface and deep-water temperature changes. These three different estimates mostly differ in the magnitude of the temperature changes but share the same overall trend (*Figure 5—figure supplement 2*). For this comparison, we only included groups overlapping the isotope record of the tree paleotemperature curves (<62.4 Myrs; resulting in 128 phylogenies).

## Diversification trends based on pulled diversification rates

To gain further insights in Neotropical diversification (Q1), we explored congruent diversification models defined in terms of pulled diversification rates (PDR, $r_p$), and pulled extinction rates (PER, $\mu_p$) (*Louca and Pennell, 2020*; *Louca et al., 2018*). Two models are congruent if they have the same $r_p$ and the same product $\rho\lambda_0$, in which $\rho$ is the sampling fraction and $\lambda_0 = \lambda(0)$. $r_p$ is equal to the net diversification rate ($r = \lambda - \mu$) whenever $\lambda$ is constant in time ($d\lambda/d\tau=0$) but differs from $r$ when $\lambda$ varies with time. The PER $\mu_p$ is equal to the extinction rate $\mu$ if $\lambda$ is time-independent but differs from $\mu$ in most other cases. Pulled and canonical diversification parameters are thus not equivalent in most cases. Biological interpretation of pulled parameters is not obvious. However, some specific properties of PDR and PER allowed us to compare diversification dynamics estimated based on pulled and canonical diversification parameters. Specifically, changes in speciation and/or extinction rates usually lead to similarly strong changes in PDR, while constant PDR are strong indicators that both $\lambda$ and $\mu$ were constant or varied only slowly over time (*Louca and Pennell, 2020*; *Louca et al., 2018*). PDR can also yield other valuable insights: if $\mu_p(0)$ is negative, this is evidence that speciation is currently decreasing over time (*Louca and Pennell, 2020*; *Louca et al., 2018*).

We estimated PDR values using the homogenous birth-death model on the R package *castor* 1.5.7 with the function *fit_hbd_pdr_on_grid* (*Louca and Doebeli, 2018*). We compared constant models (one time interval) with models in which PDR values are allowed to vary independently on a grid of three time intervals. We set up the age grid non-uniformly, for example, age points were placed closer together near the present (where information content is higher), and we selected the model that best explained the lineage-through-time of the Neotropical time trees based on AIC. To avoid non-global local optima, we performed 20 independent fitting trials starting from a random choice of model parameters. The *fit_hbd_pdr_on_grid* function additionally provided estimates of $\rho\lambda_0$ values. Knowing $\rho$, $\lambda_0$ could be derived as follows: $\lambda_0 = \lambda_0\rho/\rho$. Similarly, pulled extinction rates for each time interval could be derived as follows: $\mu p: = \lambda_0 - rp$. We limited the estimates to time periods with >10 species, using the *oldest_age* function in *castor*, to avoid points in the tree close to the root, where estimation uncertainty is generally higher.

## Neotropical bioregionalization

We used a quantitative approach to identify geographic units of long-term Neotropical evolution. We divided the Neotropical region into 13 operational areas based on the WWF biome classification (*Olson et al., 2001*) and similar to other studies, for example, (*Antonelli et al., 2018c*; *Hutter et al., 2017*) – Amazonia, Atlantic Forest, Bahama-Antilles, Caatinga, Central Andes, Cerrado, Chaco, Chocó, Guiana Shield, Mesoamerica, the Northern Andes, temperate South America, and an 'elsewhere' region – and assessed the distribution in these areas of the 12,512 species included in our 150 phylogenies. Georeferenced records were downloaded for each species through GBIF using the R package *rgbif* 0.9.9 (*Chamberlain et al., 2017*). We removed points with precision below 100 km, entries with mismatched georeference and country, duplicates, points representing country capitals or centroids, using the R package *CoordinateCleaner* 1.0-7 (*Zizka et al., 2019*). Then, we created 13 georeferenced polygons delimiting each operational area using the WWF terrestrial ecoregions annotated shapefile in QGIS, and species were assigned to each polygon according to coordinate observations using the R package *speciesgeocodeR* 1.0-4. GBIF records can result in an overestimation of widespread ranges (*Maldonado et al., 2015*), so species distributions were manually inspected for completeness and accuracy with reference to databases (AmphibiaWeb 2018, Uetz et al. 2018, GBIF.

org 2018, IUCN 2018). Based on the number of species belonging to each phylogenetic clade in the 13 ecoregions, we created a species abundance table (number of species per region per clade) that formed the basis for subsequent analyses.

The number of species distributed in each region within each clade were transformed using Hellinger transformations to account for differences in species richness between clades, and the *Morisitahorn* distance metric was selected to quantify pairwise dissimilarities of regional assemblages using the R package *vegan* 2.5-7 (**Oksanen, 2013**). We used K-means cluster analyses to form groups of similar regional assemblages. We determined the optimum number of groups by the elbow method. We use the function *fviz_cluster* in the R package *factoextra* 1.0.7 (**Kassambara and Mundt, 2017**) to visualize K-means clustering results using principal component analysis.

## Variation of diversification dynamics across taxa, environmental drivers, and biogeographic units

We classified each clade in our study according to their main taxonomic group (plant [$n=66$], mammal [$n=12$], bird [$n=32$], squamate [$n=24$], amphibian [$n=16$]), environmental correlate (as estimated above: time [$n=17$], temperature [$n=40$] or uplift [$n=17$]), species richness dynamic based on canonical diversification rates (as estimated above: Sc. 1 [$n=76$], Sc. 2 [$n=30$], Sc. 3 [$n=31$], Sc. 4 [$n=13$]), and species richness dynamic based on PDR (constant speciation [$n=83$] and decreasing speciation [$n=51$]).

We also classified each clade into the biogeographic units identified above (see results): cluster 1 (including the Amazonia, Central Andes, Chocó, Guiana Shield, Mesoamerica, and Northern Andes, [$n=97$]), cluster 2 (Atlantic Forest, Caatinga, Cerrado, Chaco, and temperate South America, [$n=10$]), cluster 3 (Bahama-Antilles, [$n=4$]), cluster 4 ('elsewhere' region, [$n=0$]), or cluster 5 (Galapagos, [$n=0$]). Clades were assigned to a given cluster only if >60% of the species appeared in the cluster, otherwise clades were classified as 'mixed' ($n=39$; **Figure 6—source data 2**).

We additionally classified clades according to the main elevational range of their constituent species following literature descriptions rather than a purely quantitative approach as for the distribution above, because GBIF records in our dataset often came without associated altitude data (<30%): lowland [<1000 m; $n=42$] including lowland rainforest in Amazonia and the Chocó in western Colombia and Ecuador, as well as rainforest in the flanking lowland and pre-montane areas along the eastern side of the Andes; montane [1000–3500 m; $n=8$] including mid-elevation montane forests (e.g., cloud and elfin forests); highland [>3500 m; $n=6$] including alpine-altitude grasslands; mixed [$n=94$] includes lineages that show a mixed preference between lowland, montane and highland. Note that in our dataset, most clades fell into the mixed category, with montane species most often occurring within clades of lowland species, and rarely forming a clade of their own. To account for this pattern (and minimize the number of clades classified as 'mixed'), we performed additional analyses pooling 'lowland' and 'montane' categories and considered a clade 'mixed' only if contained species in lowlands, montane and highlands (lowland-montane [$n=124$], highland [$n=6$], mixed [$n=20$]).

We assessed the phylogenetic signal of each multi-categorical trait (i.e., biogeographic units, elevation, diversity dynamics, and environmental drivers) using the $\delta$-statistics (**Borges et al., 2019**) over a phylogeny including one tip for each of the 150 clades represented in this study. This tree was constructed using TimeTree (**Kumar et al., 2017**). High $\delta$-value indicates strong phylogenetic signal. $\delta$ can be arbitrarily large, and thus significance was evaluated by comparing inferred $\delta$-values to the distribution of values when the trait was randomised along the phylogeny. We evaluated the phylogenetic signal of continuous traits (i.e., diversification [$r$], speciation [$\lambda$], and pulled diversification [$r_p$] rates) using Blomberg's $K$ (**Blomberg et al., 2003**) in the R package *phytools* 0.7–80 (**Revell, 2012**). Since time-varying diversification curves are hardly summarized in a single value, comparisons of net diversification values are based on estimates derived from the constant-rate model.

As no continuous or multi-categorical trait displays phylogenetic signal (see results), suggesting that the distribution of trait values is not explained by the phylogeny itself, statistical tests were conducted without applying phylogenetic corrections to account for the non-independence of data points. Fisher's exact test was used in the analysis of contingency tables, performing pairwise-comparison with corrections for multiple testing (**Benjamini and Hochberg, 1995**), and Kruskal-Wallis tests for comparing means between groups.

We also tested the effect of clade age, size, and sampling fraction on the preferred species richness dynamic (Sc. 1–4) using a phylogenetic ANOVA in *phytools* with post hoc comparisons, checking if the

residual error controlling for the main effects in the model and the tree were normally distributed. We applied phylogenetic corrections in this case because phylogenetic signal was detected for sampling fraction ($K_{sampling}$ = 0.12, p=0.001) and crown age ($K_{age}$ = 0.22, p=0.001), not for tree size ($K_{size}$ = 0.49, p=0.9).

## Acknowledgements

We are grateful to the three reviewers and the editors, for many insightful comments that helped improve the quality of our work. We thank all researchers who shared their published data through databases or with us directly (Drs Arevalo, Martins, Fortes Santos, Simon, Lohmann, Mendoza, Swenson, Erkens, van der Meijden, and Freitas). Drs J Muñoz, J Lobo, I Sanmartín, S Louca, P Manzano, and M Godefroid for invaluable comments on the manuscript and/or analyses. This work was funded by an '*Investissements d'Avenir*' grant managed by the Agence Nationale de la Recherche (CEBA, ref ANR-10-LABX-25-01) and the ANR GAARAnti project (ANR-17-CE31-0009). ASM was also supported by the *Atracción de Talento* CAM program (2019-T1/AMB-12648), Juan de la Cierva grant (IJCI-2017-32301), and Grant PID2020-120145GA-I00 funded by MCIN/AEI/ 10.13039/501100011033 and by the European Union. AA is supported by the Swedish Research Council (2019-05191), and the Royal Botanic Gardens, Kew. OAPE is funded by the Swiss Orchid Foundation and the Sainsbury Orchid Fellowship at the Royal Botanic Gardens, Kew. GC is funded by a Natural Environment Research Council Independent Research Fellowship (NE/S014470/1). RR is funded by the Spanish Ministry of Science (PID2019-108109GB-I00, AEI/FEDER). This is contribution ISEM 2022-270 of the *Institut des Sciences de l'Evolution de Montpellier*.

## Additional information

### Funding

| Funder | Grant reference number | Author |
|---|---|---|
| Agence Nationale de la Recherche | ANR-10-LABX-25-01 | Andrea S Meseguer<br>Alice Michel<br>Pierre-Henri Fabre<br>Pierre-Olivier Antoine<br>Frédéric Delsuc<br>Fabien L Condamine |
| Agence Nationale de la Recherche | ANR-17-CE31-0009 | Pierre-Henri Fabre<br>Pierre-Olivier Antoine<br>Frédéric Delsuc<br>Fabien L Condamine |
| Ministerio de Ciencia e Innovación | PID2020-120145GA-I00 | Andrea S Meseguer |
| Comunidad Autonoma de Madrid, Atraccion de Talento | 2019-T1/AMB-12648 | Andrea S Meseguer |
| Ministerio de Ciencia e Innovación | PID2019-108109GB-I00 | Ricarda Riina |
| Swedish Research Council | 2019-05191 | Alexandre Antonelli |
| Natural Environment Research Council | NE/S014470/1 | Guillaume Chomicki |
| Swiss Orchid Foundation | | Oscar A Pérez Escobar |
| Ministerio de Ciencia e Innovación | IJCI-2017-32301 | Andrea S Meseguer |

The funders had no role in study design, data collection and interpretation, or the decision to submit the work for publication.

## Author contributions
Andrea S Meseguer, Conceptualization, Resources, Data curation, Software, Formal analysis, Investigation, Methodology, Writing – original draft, Writing – review and editing; Alice Michel, Resources, Data curation, Software, Formal analysis, Writing – review and editing; Pierre-Henri Fabre, Resources, Data curation, Formal analysis, Investigation, Methodology, Writing – review and editing; Oscar A Pérez Escobar, Guillaume Chomicki, Alexandre Antonelli, Resources, Writing – review and editing; Ricarda Riina, Resources, Data curation, Investigation, Writing – review and editing; Pierre-Olivier Antoine, Resources, Funding acquisition, Writing – review and editing; Frédéric Delsuc, Conceptualization, Funding acquisition, Project administration, Writing – review and editing; Fabien L Condamine, Conceptualization, Resources, Data curation, Software, Formal analysis, Funding acquisition, Writing – original draft, Project administration, Writing – review and editing

## Author ORCIDs
Andrea S Meseguer http://orcid.org/0000-0003-0743-404X
Frédéric Delsuc http://orcid.org/0000-0002-6501-6287
Fabien L Condamine http://orcid.org/0000-0003-1673-9910

## Decision letter and Author response
Decision letter https://doi.org/10.7554/eLife.74503.sa1
Author response https://doi.org/10.7554/eLife.74503.sa2

## Additional files

### Supplementary files
• Supplementary file 1. This file contains the complete results of model selection based on (A) traditional diversification analyses comparing constant and time-dependent models; (B) traditional diversification analyses comparing constant, time-, temperature- and Andean uplift-dependent models; and (C) results for the pulled diversification rate analyses. More details are given on each table.

• Supplementary file 2. This file contains all the information on the phylogenetic reconstruction and dating of Caviomorpha.

• Transparent reporting form

### Data availability
The chronogram dataset and the diversification results are archived in Dryad. All other data used or generated in this manuscript are presented in the manuscript, or its supplementary material.

The following dataset was generated:

| Author(s) | Year | Dataset title | Dataset URL | Database and Identifier |
|---|---|---|---|---|
| Meseguer AS, Michel A, Fabre PH, Perez Escobar OA, Chomicki G, Riina R, Antonelli A, Antoine PO, Delsuc F, Condamine FL | 2021 | The Origins and Drivers of Neotropical Diversity | https://dx.doi.org/10.5061/dryad.kwh70rz4w | Dryad Digital Repository, 10.5061/dryad.kwh70rz4w |

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
