## [Editor Report]

This important work by Meseguer et al. depicts findings that substantially advance our understanding of clade diversification across major Neotropical bioregions. The evidence that summarises the evolutionary diversity dynamics of 150 time-calibrated clades of neotropical plants and animals data is convincingly presented with current state-of-the-art analyses. The work will be of interest to evolutionary biologists and biogeographers working to understand the origins of the most biodiverse land mass on the planet.

---

## [Decision Letter]

**Decision letter after peer review:**

Thank you for submitting your article "The Origins and Drivers of Neotropical Diversity" for consideration by *eLife*. Your article has been reviewed by 3 peer reviewers, and the evaluation has been overseen by a Reviewing Editor and Meredith Schuman as the Senior Editor. The following individual involved in review of your submission has agreed to reveal their identity: Dr. Yaowu Xing (Reviewer #2).

Essential revisions:

Please respond only to these essential revisions in your response letter, which have been determined through the consultative review process. In doing so you may refer to the comments in individual reviews that are related to the essential revisions.

1) We found potential inconsistencies between the title, your objectives, and your MandM. At first sight, reviewers could not see how the design of your study addresses why the Neotropics are so diverse relative to other regions, or why some regions of the Neotropics have more species than others. It is thus imperative that you temper your claims, and tone down your title.

2) Further, some reviewers expressed concern about the way divisions of the neotropics were done. Alternative splits of the Neotropical region (maybe in an appendix) could go a long way in easing concerns that traditional splits (Mesoamerica vs. Amazon vs. Andes) are not the best view of the diversity of the Neotropics.

3) Reviewers were puzzled by the lack of any confirmatory or accuracy-testing simulations on your methods. Without them, it is difficult at this point to evaluate if any strong bias exists in your results, or if your datasets have enough power to sustain your claims. It is essential that in a new version (and in your response letter) you address this issue. Does your methodology need simulations? Do currently available methods (e.g. Burin et al. 2019, Syst. Biol.; Louca and Pennell 2020, Nature) fill this gap?

*Reviewer #2 (Recommendations for the authors):*

In this study, the authors explored the evolution dynamics of Neotropical biodiversity by analyzing a very large data set, 150 phylogenies of seed plants and tetrapods. Furthermore, they compared diversification models with environment-dependent diversification models to seek potential drivers. Lastly, they evaluated the evolutionary scenarios across biogeographic regions and taxonomic groups. They found that most of the clades were supported by the expansion model and fewer were supported by saturation and declining models. The diversity dynamics do not differ across regions but differ substantially across taxa. The data set they compared is impressive and comprehensive, and the analysis is rigorous. The results broadened our understanding of the evolutionary history of the Neotropical biodiversity which is the richest in the world. It will attract broad interest to evolutionary biologists as well as the public who are interested in biodiversity.

The paper is well and clearly written in general.

1. My concern is about the sampling. It seems the authors only sampled the species that occurred in the Neotropics. It is OK for clades mostly distributed in this region. On the other hand, the sampling strategy overlooked the role of dispersal in generating biodiversity of Neotropics. However, many studies have already shown that dispersal (including long-distance dispersal) played important role in shaping Neotropical biodiversity. I suggest the authors discuss this somewhere in the Discussion part. Furthermore, I noticed that many tree sizes are small with less than 10 species. I do not know how reliable to estimate diversification rates for small-sized trees.

2. Though the authors quantified the effect of climate change, it seems the current resolution can not capture the role of Pleistocene climatic fluctuations which was proven to be important for particular biomes such as the alpine Paramos. Adding some discussions will strengthen the conclusions.

3. The first paragraph of the Conclusion part mainly illustrated the limitations of the study. I suggest the authors could put this part in the Discussions.

*Reviewer #3 (Recommendations for the authors):*

I was very impressed by the scope of the dataset, the thoroughness of the analyses, and the attention paid to caveats and limitations. I've provided a few comments below that are intended to strengthen this exciting manuscript.

One point I thought that might be of use to mention is that the focus on the recent past (very late Mesozoic-Cenozoic) means that South America's position relative to the present has varied little (ie. it was pretty much already in place by the time the study period begins), and this has implications of how much the climate has changed in these equatorial regions relative to more temperate regions (and the onset of the LDG during this time as global temps cooled).

L340-345: Were the smallest clades excluded also generally the youngest?

L356-357: Was this due to differences in clade size/age?

L364-375: Any idea if the discordance in endotherm diversification dynamics was due to New World monkeys vs. all other endotherms occupying a different range of latitudes? I'm just thinking about how lineages occupying more equatorial latitudes may have experienced a more limited drop in temperatures through time, relative to those at higher latitudes, and am wondering if this geographic separation (if it exists) could potentially explain the observed discordance?

L384-386: It might be useful to note here that while this may be the case, terrestrial tetrapods haven't gone completely extinct, as they've persisted since the Carboniferous.

L402-408: I like that attention is paid to potential biotic drivers of plant diversification; however, I'm also left wondering what role the evolution of angiosperm-dominated rainforests (which changed both the climate and vegetation structure) played any role in shaping the evolution of terrestrial tetrapods. I know it is beyond the scope of this paper to formally address it, but I think briefly mentioning the plausibility of this as a potential explanation for time-variable diversification dynamics would be beneficial.

L411-412: Any idea if this is due to the absence of other large clades restricted to the Neotropics, a lack of phylogenies for these, or because neotropical species are embedded in clades that have substantially radiated elsewhere (or some combination of these)?

L438: I think it's important to remember that these biomes, ecoregions and biogeographic units may be relatively recent.

L449-452: How can species richness patterns identify shared evolutionary histories? It feels as though it might be challenging to disentangle whether these richness patterns reflect in situ diversification in that biome or dispersal into that biome.

L498: uncomplete -> incomplete

L502: divers -> diverse

L524-531: Please clarify if any occurrence (including a single occurrence when 99% of occurrences are from outside the neotropics) was enough to code the taxon as occurring in the Neotropics, or if a higher threshold was set (xx% of occurrences from Neotropics). And the same request for assigning to taxa to the 13 WWF biomes.

L685-687: Was the 60% criterion applied above used here to determine whether clades were scored as mixed, lowland, montane, or highland?

L699-700: Doesn't this show that there is NOT phylogenetic signal (as it is written, it states that any trait contains phylogenetic signal, even though the K values are low and non-significant).

[Editors’ note: the authors submitted for reconsideration following the decision after peer review. What follows is the decision letter after the second round of review.]

Thank you for resubmitting the paper entitled "Diversification dynamics in the Neotropics through time, clades and biogeographic regions" for further consideration by *eLife*. Your revised article has been evaluated by a Senior Editor and a Reviewing Editor. We are sorry to say that we have decided that this submission will not be considered further for publication by *eLife*.

Both Reviewers and we (editors) valued the work done in the revision, but further comments raised by Reviewer 1 further weakened the validity of your approach. Specifically, the lack of a controlled methodology (by confirmatory or accuracy-testing simulations) made it difficult to evaluate if any strong bias exists in your results, or if your datasets have enough power to sustain your claims.

*Reviewer #1 (Recommendations for the authors):*

I previously reviewed this manuscript in December of 2021. I had several concerns. One was that the main method used to estimate diversification over time might be problematic. Another was that there was little comparison among different regions to understand patterns of diversity among regions in the Neotropics. I appreciate that the authors have made efforts to address these concerns, but I think that these problems remain somewhat problematic.

First, despite what the authors claim, I think the main approach used has not really been thoroughly tested with simulations. The authors claim that it has been, specifically by Morlon et al. (2011), Lewitus and Morlon (2018), and Condamine et al. (2019). The paper by Morlon et al. (2011) contains simulations, but (as far as I know) those simulations do not actually address the ability of the method to accurately choose among different models of diversification over time. The paper by Condamine et al. (2019) does not contain simulations at all. The paper by Lewitus and Morlon (2018) contains simulations, but primarily compares the ability of the method to distinguish time-dependent and temperature-dependent models (their Figure 3). But it is unclear if these two classes of models can be accurately distinguished from cases where the true model is a constant-rate model. Furthermore, looking at Figure 3 of Lewitus and Morlon (2018) it can be seen that this method often selects a constant-rate model when the true model is temperature dependent, if the effect of temperature dependence on diversification over time is not strong enough. The authors include in this list only papers co-authored by the developer of the method, and not other papers that found the method to be more problematic. It is also notable that the diversification dynamics are not significantly different between plants and animals when using PDR (Table 2). Therefore, not all of the main results are concordant between these two methods.

I found the comparison among biogeographic regions to be unsatisfying overall, because the regions are extremely coarse grained and each incorporates so many different regions. Furthermore, the comparison of 97 clades in one region (Mesoamerica+Amazonia+Andes and others) vs. only 10 clades in another (southern South America) also seems unsatisfying.

I appreciate that the authors describe their four scenarios explicitly in the Introduction, but I do not think that knowing the frequency of these four scenarios is "required to understand the origin and maintenance of Neotropical diversity" (line 112). Also, I do not see why the accumulation of species through "pulses" corresponds to the exponential expansion scenario. Should not constant speciation and low, constant extinction lead to an exponential increase in diversity over time? Even in Figure 1, I do not see the pulses in Scenario 2. A "pulse" implies a pattern that is discontinuous or episodic over time.

The finding that diversification patterns are different between plants and animals is interesting, but not entirely novel (see for example, Hernandez-Hernandez et al. 2021; Biological Reviews on the faster rates of diversification in plants vs. animals). It is also notable (again) that the diversification dynamics over time are not significantly different between plants and animals when using PDR, even if the rates are.

Finally, I am not sure how relevant these results are to future climate change, as is stated in the Abstract. It seems like this idea is not addressed in the paper itself.

Specific comments:

Lines 285 to 287: This needs to be rewritten since the results seem to contradict the conclusions.

Lines 340-342. The study cited focused on BiSSE-type models, not the approach used here.

Line 374. Change "plant phylogenies are significantly worst sampled than those of most other tetrapods" to "plant phylogenies are significantly less sampled than are tetrapod phylogenies"

Lines 415 to 416. How can there be microevolutionary studies of diversification?

Lines 538 to 540. The paper by Kreft and Jetz does not address diversification, only present-day species richness.

Lines 554 to 555. The "diverse range of diversification methods compared here" is actually only two diversification methods, by my count.

Line 563. What "substantial proportion of Neotropical diversity" is accounted for by these models?

*Reviewer #2 (Recommendations for the authors):*

The authors have addressed all of my comments. I am happy with the current version.

*Reviewer #3 (Recommendations for the authors):*

I thank the authors for their responses to my previous queries and approve of the changes made.

[Editors’ note: further revisions were suggested prior to acceptance, as described below.]

Thank you for choosing to send your work entitled "Diversification dynamics in the Neotropics through time, clades and biogeographic regions" for consideration at *eLife*. Your letter of appeal has been considered by a Senior Editor and a Reviewing Editor, and we are prepared to consider a revised submission with no guarantees of acceptance.

In this revised version, we stress that you must carefully address the following: (1) Please be more cautious about reporting the constraints of your chosen methods, (2) Please dilute the claim that you are resolving South American diversification, (3) Please address the several aspects of the most recent reviews which were not addressed in your appeal letter, including (but not limited to): your reliance on one specific method to calculate diversification rates, an apparent lack of simulations to support your chosen method, the coarse-grained resolution of the study and imbalance in clade numbers investigated in different regions, the request for a clearer and more critical projection of scenarios resulting from the proposed speciation patterns (do pulses necessarily lead to exponential expansion), and being careful to base conclusions about diversification patterns on dynamics and rates.

---

## [Author Response]

Essential revisions:Please respond only to these essential revisions in your response letter, which have been determined through the consultative review process. In doing so you may refer to the comments in individual reviews that are related to the essential revisions.1) We found potential inconsistencies between the title, your objectives, and your MandM. At first sight, reviewers could not see how the design of your study addresses why the Neotropics are so diverse relative to other regions, or why some regions of the Neotropics have more species than others. It is thus imperative that you temper your claims, and tone down your title.

We thank you for your insightful comment and welcome these suggestions. We realize that the title we had initially chosen was so general that it might not reflect the content of the article. In our study we have four objectives: (1) to describe the main macroevolutionary dynamics that have prevailed in the Neotropics through time at a continental scale; (2) to assess their drivers; and (3) how they varied across biogeographic units and *(4)* taxonomic groups.

We agree with the editor and reviewers that we cannot directly answer why the Neotropics are more diverse than other regions of the world. To this end, we would need to compare our results (on the prevailing Neotropical dynamics) with comparative studies from other regions. We have changed the title of the study in the revised version as follows: “Diversification dynamics of plants and tetrapods in the Neotropics through time, clades and biogeographical regions”. We hope you will find this new title a better one reflecting the content of the article. In addition, to avoid any confusion, we have deleted the following sentence from the introduction: “But such an assessment is required to understand the origin of Neotropical diversity and why the Neotropics are more diverse than other regions in the world”.

Nevertheless, we would argue that our study could partially shed light on why some Neotropical regions have more species than others. Contrasted diversity levels across regions may be explained by three non-exclusive hypotheses: (1) species richness of a given region might be correlated to the amount of time available for speciation ("time-to-speciation" hypothesis; Stephens and Wiens, 2003); (2) some regions may just accumulate species at higher rates (diversification rate hypothesis); and (3) some regions may have more species because of higher rates of incoming dispersal (dispersal hypothesis). Assessing the relative support of these factors is not the main goal of our study, but our results allow us to evaluate the diversification rate hypothesis. We assess whether diversification rates vary across biogeographic units, and found similar diversification patterns and rates among regions. Thus, our results do not support the diversification rate hypothesis, and suggest that other factors might be at play. We have included this additional rationale in our Discussion: “Our results therefore suggest that differences in species richness between the Neotropical bioregions identified here might not be attributable to differences in diversification rates, and that other factors such as time (127), or asymmetric dispersal (4) could explain this pattern”.

2) Further, some reviewers expressed concern about the way divisions of the neotropics were done. Alternative splits of the Neotropical region (maybe in an appendix) could go a long way in easing concerns that traditional splits (Mesoamerica vs. Amazon vs. Andes) are not the best view of the diversity of the Neotropics.

Thank you for this comment. We have to admit that the division of the Neotropics into areas reflecting clade (and not species) endemicity has been the most complicated part of the study. There is a lot of discussion in the literature on the bioregionalization of species ranges and no consensus (Antonelli et al., 2018 – PNAS). When it comes to the bioregionalization of clade ranges, there is very few information and harder decisions to make. We started coding each phylogeny following classical bio-regionalization schemes (e.g., biomes, ecoregions), but quickly realized that they were not useful at the scale of this study, since most clades have species distributed in most Neotropical ecoregions. Thus, for previous versions of this manuscript, we decided to categorize clades within the Neotropics based on the three dominant patterns described by Gentry (1982 – Annals Miss. Bot. Gar.) that roughly characterize the main distribution of species richness, as well as the colonization history of Neotropical lineages: *Andean-centered* = lineages with species richness mostly distributed in the Andes, Chocó and Central America, and poorly represented in Amazonia, Atlantic Forest, Caatinga and Cerrado; *Amazonian-centered* = species richness mostly distributed in the Amazonia and Brazilian shield and poorly represented in Central America; and *other* = distribution that does not match the two previous categories (*e.g.* Southern cone of South America). We then compared the diversification dynamics between these regions and found similar results to those presented in this manuscript: the prevailing diversification dynamics and rates do not change between Gentry’s regions (see Author response image 1).

**Author response image 1. sa2fig1:** Variation in diversification rates on 150 Neotropical phylogenies of plants and tetrapods across geographic ranges. Diversification rates are derived from the constant-rate model.

However, as could be noticed, this classification is quite subjective: “… lineages with species richness MOSTLY or POORLY distributed in this or the other area” are rather subjective terms, which could lead other researchers to apply different criteria to categorize the same phylogenetic tree. Other traditional splits, as suggested here into Mesoamerica vs. Amazon vs. Andes, are even more subjective. How should we classify clades with species distributed in the Chocó, or in the Cerrado, for example, under this scheme?

Therefore, in this study, we decided to reduce all subjective divisions and use a quantitative approach to identify macroevolutionary Neotropical bioregions. We think that this choice is one of the greatest strengths of our study, as it makes our results repeatable and more objective than previously. In addition, by using this quantitative approach, we have revealed new patterns not identified before: for example, we have found that the Andes, Amazonia, and Chocó have acted as a single geographic unit of biotic evolution at the clade level for tens of millions of years. The existence of regional arenas of evolution in the Neotropics at the clade level and at large time scales is undoubtedly a key finding of this study. If the editor agrees, we would prefer not to include alternative and less objective divisions of the Neotropics in the study, which, as we demonstrate with our study, do not reflect the endemism patterns of Neotropical clades, and might thus lead to confusion.

3) Reviewers were puzzled by the lack of any confirmatory or accuracy-testing simulations on your methods. Without them, it is difficult at this point to evaluate if any strong bias exists in your results, or if your datasets have enough power to sustain your claims. It is essential that in a new version (and in your response letter) you address this issue. Does your methodology need simulations? Do currently available methods (e.g. Burin et al. 2019, Syst. Biol.; Louca and Pennell 2020, Nature) fill this gap?

This is a legitimate comment, and we understand the skepticism on a study that relies on macroevolutionary models of questionable robustness (e.g. Kubo and Iwasa 1995 – Evolution; Rabosky and Lovette 2008 – Evolution; Crisp and Cook 2009 – Evolution; Quental and Marshall 2010 – TREE; Burin et al. 2019 – Syst. Biol.; Louca and Pennell 2020 – Nature; Pannetier et al. 2021 – Evolution).

The methodology used here has been thoroughly tested with both simulations (e.g. Morlon et al. 2011 – PNAS; Lewitus and Morlon 2018 – Syst. Biol.; Condamine et al. 2019 – Ecol. Lett.) and empirical cases (e.g. Lewitus et al. 2018 – Nat. Ecol. Evol.; Condamine et al. 2019 – Ecol. Lett.). We cannot deny that such a methodology is fully free from issues, which affect all birth-death models, and brings the question: are we able to reliably infer the diversification model and identify parameter values of this model (Louca and Pennell 2020 – Nature)? These concerns are not likely to be resolved in the short term. Although many studies are making progress in understanding the behavior of diversification rate functions, showing, for example, that equally likely diversification functions (i.e. the congruent parameter space of Louca and Pennell 2020 – Nature) can share common features, with diversification rate patterns being robust despite non-identifiability (Höhna et al., 2022 – bioRxiv; Morlon et al., 2022 – TREE).

Being aware of these concerns, we also relied on the recently developed Pulled Diversification Rates method (Louca and Pennell 2020 – Nature; Louca et al., 2018 – PNAS) that is supposed to correct for the identifiability issue raised by recent studies. Hence, applying both traditional and pulled birth-death models to all phylogenies, we have shown a good consistency in the inferred models, which suggests that our study can provide meaningful estimates of diversification. Our empirical study is also one of the first to perform such a large-scale methodological comparison in diversification analyses (pulled vs. traditional birth-death models) while addressing a key question in evolutionary biology. We have now emphasized this point in the conclusions of our study: “To the extent possible, these results are based on traditional diversification rates, and on the recently developed Pulled Diversification Rates method that is supposed to correct for the identifiability issue raised by recent studies associated with traditional diversification rates (71). Hence, applying both traditional and pulled birth-death models to all phylogenies, we have shown a good consistency in the inferred models, which suggests that our study can provide meaningful estimates of diversification”.

Reviewer #2 (Recommendations for the authors):In this study, the authors explored the evolution dynamics of Neotropical biodiversity by analyzing a very large data set, 150 phylogenies of seed plants and tetrapods. Furthermore, they compared diversification models with environment-dependent diversification models to seek potential drivers. Lastly, they evaluated the evolutionary scenarios across biogeographic regions and taxonomic groups. They found that most of the clades were supported by the expansion model and fewer were supported by saturation and declining models. The diversity dynamics do not differ across regions but differ substantially across taxa. The data set they compared is impressive and comprehensive, and the analysis is rigorous. The results broadened our understanding of the evolutionary history of the Neotropical biodiversity which is the richest in the world. It will attract broad interest to evolutionary biologists as well as the public who are interested in biodiversity.The paper is well and clearly written in general.1. My concern is about the sampling. It seems the authors only sampled the species that occurred in the Neotropics. It is OK for clades mostly distributed in this region. On the other hand, the sampling strategy overlooked the role of dispersal in generating biodiversity of Neotropics. However, many studies have already shown that dispersal (including long-distance dispersal) played important role in shaping Neotropical biodiversity. I suggest the authors discuss this somewhere in the Discussion part. Furthermore, I noticed that many tree sizes are small with less than 10 species. I do not know how reliable to estimate diversification rates for small-sized trees.

Thank you for pointing this out. Our sampling indeed makes a focus on clades endemic to the Neotropics, i.e. independent Neotropical radiations, which by definition includes *in-situ* diversification and limits the role of dispersal into the Neotropics. This procedure ensures that the diversification signal we analyze pertains to the Neotropics and not to other areas. We do not neglect dispersal as a driver, but for that we would need to rely on a biogeographical approach that is beyond the scope of this study. In the revised manuscript, we have acknowledged that dispersal into and out of the Neotropics is an additional factor that we did not take into account in this study. We have added the following sentence in the discussion: “Furthermore, by focusing exclusively on Neotropical radiations, we did not consider the role of dispersal into and out of the Neotropics (or within Neotropical regions) as an additional factor explaining Neotropical diversity. In fact, several studies have suggested an important role of dispersal in the configuration of modern Neotropical biotas (Smith et al., 2014; Carrillo et al., 2020; Bacon et al., 2015). Future studies integrating biogeographic and diversification processes will be needed to bridge this gap, and to provide a more complete picture on the drivers of Neotropical diversity.” Please see also above Essential revision 1.

Estimating rates of diversification for small-sized trees is undeniably difficult and comes with more uncertainties. However, we wanted to include these small phylogenies because they also represent the Neotropical biodiversity. We did not want to introduce a bias in our meta-analysis toward large-sized trees that would only show variable-rate models and would hide a substantial part of the biodiversity associated with this diversification pattern. We also show that constant-rate models can explain the diversification in the Neotropics. There is a more balanced diversification pattern, and we think it is fairer to present both large and small phylogenies. Nevertheless, we agree that small trees are problematic. We have evaluated the effect of including small trees in our sample. As shown in our Figure 9, clades fitting gradual expansion models tend to have less species than clades fitting exponential (*p*=0.006) and declining (*p*=0.03) dynamics, suggesting that small trees could introduce a bias in our results. We therefore repeated all the analyses excluding small trees (<20% of the species sampled) from our analyses, and found that the largest support for the expanding diversity trend persisted (72–60% of clades).

We discuss the difficulty to estimate rates of diversification from small-sized phylogenies and the possible bias introduced by the small phylogenies in our study in the Discussion section: “It has been suggested that the power of birth–death models to detect rate variation can decrease with the number of species in a phylogeny (Davis et al., 2013 – BMC Evol. Biol.), suggesting that tree size could hinder the finding of rate-variable patterns. Still, the patterns found in this study appear to be robust to sampling artifacts. We have repeated all the analyses excluding small (<20 species) and poorly sampled (<20% of the species sampled) phylogenies from our analyses, and found that the largest support for the expanding diversity trend persisted (72–60% of clades). Then, the relative support for the exponentially expanding scenario (Sc. 2) increased at the expense of the gradually expanding scenario (Sc. 1), strengthening the conclusion of the generality of the expanding trend in the Neotropics”. We hope you will find this result sufficient to demonstrate that the consideration of small trees in our analyses does not drive our main conclusions.

2. Though the authors quantified the effect of climate change, it seems the current resolution can not capture the role of Pleistocene climatic fluctuations which was proven to be important for particular biomes such as the alpine Paramos. Adding some discussions will strengthen the conclusions.

Thank you for this comment. We agree that Pleistocene climatic changes have had notable effects on speciation and extinction around the world. This has been largely studied and documented, and our study aims at extending the time frame without ignoring these findings. First of all, note that the temperature data we used includes the Pleistocene climatic fluctuations (e.g. Zachos et al. 2001 – Science; Zachos et al. 2008 – Nature), but it is true that smoothing the temperature curve tends to erase the important features of the Pleistocene fluctuations. In addition, the birth-death models used here cannot fit well these fine-scale variations because there are a lot of temperature ups and downs over a short time interval, while the number of speciation events in the same time frame is often low. We think it is probably illusory to believe that macroevolutionary models can provide reliable inferences for microevolutionary processes. Our study aims at providing estimates on a longer time-scale, which is often lacking compared to microevolutionary studies that have yielded numerous examples of the effect of Pleistocene climatic changes on biodiversity. That being said, we have revised the manuscript to discuss the role of recent climatic changes. We have included the following sentences: “The role of recent climate change over Neotropical diversity has been widely documented, especially concerning the impact of Pleistocene climatic fluctuations on tropical forests (Haffer 1968; Rull 2011), or in the formation of species-rich biomes such as the Paramo (Madriñan et al., 2019). Recent climatic variations are probably not sufficiently captured in our study. Our diversification models cannot fit well the fine-scale variations of the Pleistocene fluctuations because there were a lot of temperature ups and downs over a short time interval, while the number of speciation events in the same time frame can be low. In addition, the temperature curve is smoothed when incorporated into the model, which probably erases some of the important features of these fluctuations. Our study aims at providing estimates on a longer time-scale, which is often lacking in microevolutionary studies. Although Neotropical climate has been relatively stable through the Cenozoic in comparison to other regions (94, 95) – in the Neotropics global cooling contributed to the expansion of several biomes, such as the alpine Paramos (62) or open ecosystems (96, 97) – temperature changes emerge in our study as an important factor driving Neotropical diversity across macroevolutionary scales. These results are in line with previous studies suggesting that temperature is a key driver of biodiversity change at different evolutionary scales (7, 43). However, our results also reveal that climate change is not the only factor shaping Neotropical diversity at macroevolutionary scales.”

3. The first paragraph of the Conclusion part mainly illustrated the limitations of the study. I suggest the authors could put this part in the Discussions.

Thank you for this suggestion. We agree and have revised the main text accordingly.

Reviewer #3 (Recommendations for the authors):I was very impressed by the scope of the dataset, the thoroughness of the analyses, and the attention paid to caveats and limitations. I've provided a few comments below that are intended to strengthen this exciting manuscript.

Thank you for your review, the positive input and all the comments.

One point I thought that might be of use to mention is that the focus on the recent past (very late Mesozoic-Cenozoic) means that South America's position relative to the present has varied little (ie. it was pretty much already in place by the time the study period begins), and this has implications of how much the climate has changed in these equatorial regions relative to more temperate regions (and the onset of the LDG during this time as global temps cooled).

This is an important point to make. Indeed, the timeframe of the study is mostly focused on a period when South America was approximately at the same place and latitudes as today. It broke up from Africa around 110 million years ago (e.g. Seton et al. 2012 – Earth-Sci. Rev.; Müller et al. 2016 – Annu. Rev. Earth Planet. Sci.), and slowly moved to reach its current position. However, we don't think this factor explains the pattern found here, as the latitudinal position of other continental landmasses also varied little during the Cenozoic. Except for Africa, which collided with the Arabian plate during the Miocene, North America and Eurasia were also pretty much in the same latitudinal position during the entire Cenozoic (Sanmartin et al. 2001 – Biol. J. Linn. Soc.; Seton et al. 2012 – Earth-Sci. Rev.; Müller et al. 2016 – Annu. Rev. Earth Planet. Sci.). It is true that global Cenozoic cooling was much less intensively felt in South America than in other regions, and the region remained tropical since the beginning of the Cenozoic, although global cooling contributed to the expansion of more open ecosystems (Cheng et al. 2013 – Nat. Com.; Dick and Pennington 2019 – Annu. Rev. Ecol. Evol. Syst.). In the revised version of the main text, we have mentioned this aspect pertaining to South America, its tropicality, and the effect of climate change at tropical latitudes:

“The role of recent climate change over Neotropical diversity has been widely documented, especially concerning the impact of Pleistocene climatic fluctuations (Haffer 1968; Rull 2011) […] Our study aims at providing estimates on a longer time-scale, which is often lacking in microevolutionary studies. Although Neotropical climate has been relatively stable through the Cenozoic in comparison to other regions (94, 95) – in the Neotropics global cooling contributed to the expansion of several biomes, such as the alpine Paramos (62) or open ecosystems (96, 97) – temperature changes emerge in our study as an important factor driving Neotropical diversity across macroevolutionary scales. These results are in line with previous studies suggesting that temperature is a key driver of biodiversity change at different evolutionary scales (7, 43)”.

L340-345: Were the smallest clades excluded also generally the youngest?

We did not observe any relationship between clade age and number of species. Models assuming constant speciation and extinction rates through time best fit half of the phylogenies in our study, often when clades are species-poor. Phylogenies best fitting a constant model are also significantly younger. However, young clades do not tend to be species-poor (Pearson's r = 0.35; see Author response image 2). We did not find a correlation between the other variables either; crown age – sampling fraction (r = 0.15), sampling – number of tips (r = 0.11).

**Author response image 2. sa2fig2:** Scatterplots showing the relationship between (log transformed) sampling fraction, clade age (crown age in million years ago), and number of tips for the 150 phylogenies examined in this study. The plots shows no association between the variables.

L356-357: Was this due to differences in clade size/age?

This is a very good point. We agree this comparison is relevant to support our conclusions, but it was missing from our results. We have now compared tree size, crown age and sampling fraction across taxonomic groups, and found that the higher proportion of increasing dynamics, characteristic of plants, cannot be explained by significant differences in these factors. As can be seen on the figure below (new Figure-2—figure supplement 2 on the manuscript), tree size does not differ among plants, mammals, birds and squamates. Crown age does not differ among plants, mammals and birds. Groups do differ on sampling fraction: plant (p < 0.01) and squamate (p < 0) phylogenies are significantly worst sampled than the phylogenies of other groups. Yet plants show a higher frequency of increasing dynamics than squamates, and other tetrapods (Figure 4). Incomplete taxon sampling has the effect of flattening out lineages-through-time plots towards the present, and thus artificially increasing the detection of diversification slowdowns rather than diversification increases (Cusimano and Renner 2010 – Syst. Biol.).

We have included this important piece of information in the results “In our dataset, amphibian phylogenies are significantly larger than those of other clades (p < 0.05) (Figure 2 —figure supplement 2). Amphibian and squamate phylogenies are also significantly older (p < 0). Groups also differ in sampling fraction: plant (p < 0.01) and squamate (p < 0) phylogenies are significantly worst sampled than phylogenies of other groups.”; and in the Discussion section: “Differences in the phylogenetic composition of the plant and tetrapod datasets do not explain this contrasted pattern. On average, plant phylogenies are not significantly younger or species-poorer than tetrapod phylogenies (Figure 2 —figure supplement 2). Yet, the proportion of clades experiencing increasing dynamics is significantly higher for plants (Figure 4). Plant phylogenies are significantly worst sampled than those of most other tetrapods, though, as explained above, incomplete taxon sampling has the opposite effect: flattening out lineages-through-time plots towards the present (83).”

L364-375: Any idea if the discordance in endotherm diversification dynamics was due to New World monkeys vs. all other endotherms occupying a different range of latitudes? I'm just thinking about how lineages occupying more equatorial latitudes may have experienced a more limited drop in temperatures through time, relative to those at higher latitudes, and am wondering if this geographic separation (if it exists) could potentially explain the observed discordance?

Very interesting point. A priori, our data do not suggest a contrast in the latitudinal distribution of New World monkeys vs. other endotherms to explain this difference. Most species of New World monkeys appear distributed in tropical latitudes, and this clade was assigned to our biogeographic cluster 1 (the “Pan-Amazonia” macroregion). We quantified that most species of monkeys occur in the Amazonia (61 spp.), Northern Andes (25), Guiana Shield (17), Central Andes (14), and Chocó (12) (Figure 6 – Source data 1). Some species also occur in other areas, such as the Atlantic Forest (14 spp.), Cerrado (12). Very few species are distributed across temperate latitudes: we only identified 8 species “Elsewhere”, 5 species occur in the Chaco region, and none in temperate South America. This distribution pattern is the most common in our mammal dataset (Figure 6 – Source data 1 and 2). Therefore, we think the contrasted diversification dynamics found here might be real, reflecting a better ability of New World monkeys to adapt to the changing conditions that occurred in South America during the Cenozoic, or be an artifact of taxonomic over-splitting of species in this clade, given that previous studies have suggested that the number of species has traditionally been inflated in this clade (see Springer *et al.* 2012 – PLoS One for a discussion on this taxonomic issue).

L384-386: It might be useful to note here that while this may be the case, terrestrial tetrapods haven't gone completely extinct, as they've persisted since the Carboniferous.

We agree, and we did not mean that terrestrial tetrapods have gone extinct but rather that they have experienced some periods of slowdown in diversification rates, or even periods of diversity decline (where the number of species has decreased) but eventually bounced back. We have clarified this sentence as follows: “Our results lend support to an alternative explanation for diversification slowdowns: the idea that tetrapods, for some periods, were less successful in keeping pace with a changing environment.”

L402-408: I like that attention is paid to potential biotic drivers of plant diversification; however, I'm also left wondering what role the evolution of angiosperm-dominated rainforests (which changed both the climate and vegetation structure) played any role in shaping the evolution of terrestrial tetrapods. I know it is beyond the scope of this paper to formally address it, but I think briefly mentioning the plausibility of this as a potential explanation for time-variable diversification dynamics would be beneficial.

This is a relevant point. We acknowledge that the evolution of angiosperm-dominated rainforests played an important macroevolutionary role in shaping the evolution of terrestrial biodiversity. Some of us have studied its role over the diversification of conifers, which have likely been outcompeted by angiosperms (e.g. Condamine et al. 2020 – PNAS). However, we don't think the evolution of an angiosperm-dominated forest significantly affected diversification dynamics of tetrapods in our study. It is considered that angiosperm-dominated forests were already established in the Neotropics by the Paleocene (Johnson and Ellis 2002 – Science; Wing et al. 2009 – PNAS; Carvalho et al. 2021 – Science), while the age of origin for most clades in our study postdates this period (Figure 2). It is possible that this biome transition affected the early evolution of a few clades of frogs and squamates, whose crown ages date back to the end of the Cretaceous. However, most cladogenetic events on these clades postdate the Paleocene. In the revised manuscript, we have incorporated the fact that our model design is limited to the tested hypotheses and that there could have been other untested variables, such as the role of angiosperm-dominated rainforests, that could drive Neotropical diversification: “Similarly, we did not assess the role of the emergence of angiosperm-dominated rainforests in the evolution of tetrapods. Angiosperm-dominated forests were already established in the Neotropics by the Palaeocene (104), while the age of origin for most clades in our study postdates this period (Figure 2)”.

L411-412: Any idea if this is due to the absence of other large clades restricted to the Neotropics, a lack of phylogenies for these, or because neotropical species are embedded in clades that have substantially radiated elsewhere (or some combination of these)?

An interesting point that is somewhat difficult to address. If we have to pick one of the three proposals, we would bet on the incomplete representation in phylogenies of species from large plant genera or from subclades within those genera. Most megadiverse plant genera (> 500 spp.) are well represented in the Neotropics (e.g., *Anthurium, Astragalus, Begonia, Carex, Croton, Epidendrum, Eugenia, Lepanthes, Masdevallia, Maxillaria, Miconia, Mimosa, Myrcia, Passiflora, Peperomia, Piper, Pleurothallis, Psychotria, Salvia, Solanum, Tillandsia;* Ulloa et al. 2017 – Science). Although there are phylogenetic studies including species from all of them, the sampling is largely incomplete in most cases. In general, phylogenetic studies of large groups can only afford to include a small fraction of the species, although they try to represent the morphological/geographical diversity of the group. A few examples of groups we are familiar with that still suffer from incomplete phylogenetic knowledge in the Neotropics are: *Brongniartia, Dalbergia, Harpalyce, Zapoteca* (Leguminosae); *Acalypha*, *Croton* sect. *Adenophylli*, *Jatropha*, *Mabea*, *Sapium, Sebastiania* (Euphorbiaceae); *Eugenia, Myrcia*, *Myrcianthes* (Myrtaceae); *Arachnothryx, Palicourea, Psychotria* (Rubiaceae); *Ficus* (Moraceae), *Navia* (Bromeliaceae); *Abolboda*, *Xyris* (Xyridaceae), *Paepalanthus* (Eriocaulaceae); *Epidendrum* (Orchidaceae).

Another issue to consider is that we are missing data from phylogenetic studies that were published after the closure of our data collection (2018–2022). It is inevitable to have a gap in the sampling due to the time lapse since data gathering and the actual publication of the study.

Finally, it is true that there are Neotropical species/groups that are embedded in larger clades, which are more species-rich elsewhere. For these cases, it is not possible nor relevant to study the diversification history in the Neotropics, because the Neotropical component in those groups is represented by a handful of species.

L438: I think it's important to remember that these biomes, ecoregions and biogeographic units may be relatively recent.

We agree. This is a very good point. Thank you for bringing it up. There is a body of evidence for the age of some of these biomes, which indeed indicates a recent age for some. For instance, the Cerrado is considered to have originated 9 million years ago (Simon et al. 2009 – PNAS). The Chocó region is supposed to be of more recent origin (Pérez-Escobar et al. 2019 – Front. Plant Sci.). This is relevant to explain why traditional bio-regionalization schemes are not appropriate to describe geographic patterns at the scale of this study. In the revised manuscript, we have mentioned this relevant point: “Often, these bioregions have been shown to be useful for categorizing actual species ranges, but they are less appropriate for examining endemism at the macroevolutionary scale. The age of origin of many of these bioregions postdates the origin of many of our clades (Figure 2). For instance, the Cerrado originated during the late Miocene (110), and the Chocó during the Pliocene-Pleistocene (111). In addition, most clades in our study appear distributed in most Neotropical ecoregions (Figure 6 – Source Data 1)”.

L449-452: How can species richness patterns identify shared evolutionary histories? It feels as though it might be challenging to disentangle whether these richness patterns reflect in situ diversification in that biome or dispersal into that biome.

The biogeographic clusters or macroregions identified here reflect areas of endemism at the clade level. These regions have been identified based on different clades sharing a similar pattern, in which most of the clade's species occur within the macroregion, and very few in other regions. This suggests that the contribution of *in situ* diversification might be more relevant than dispersal to explain the biotic assemblage of these macroregions. Accordingly, we initially defined these macroregions as regions of "shared evolutionary history". However, we realize that this term can be confusing and in the revised version of the manuscript we have rephrased it to clarify the meaning of these macroregions: “We propose here an alternative bioregionalization scheme of the Neotropical region that accounts for long-term regional assemblages at macroevolutionary scales (Figure 6). We identify five biogeographic units that represent macroregions where different independent Neotropical radiations occurred over millions of years of biotic evolution. These regions are defined in terms of species-richness patterns within clades (Figure 6 – Source Data 1; Figure 6 – Source Data 2), showing that species-rich clades in Amazonia, also tend to be species-rich in the Andes, Chocó, Guiana Shield, and Mesoamerica (biogeographic cluster 1), without excluding that some species within these clades occur in other regions…[]…As such, these biogeographic clusters form distinctive units of Neotropical evolution and represent long-term clade endemism. Within each of these clusters, the contribution of in situ diversification is therefore more relevant than dispersion to explain their biotic assemblage”.

L524-531: Please clarify if any occurrence (including a single occurrence when 99% of occurrences are from outside the neotropics) was enough to code the taxon as occurring in the Neotropics, or if a higher threshold was set (xx% of occurrences from Neotropics). And the same request for assigning to taxa to the 13 WWF biomes.

Yes, in our approach, we considered that a single occurrence was enough to assign a taxon to a given region. We are aware that this approach has limitations, especially since GBIF data often present inaccurate or erroneous occurrences (Maldonado et al. 2015 – Glob. Ecol. Biogeogr.). However, it is paramount to notice that, from the raw occurrence dataset, we removed occurrences with precision below 100 km, entries with mismatched georeference and country, duplicates, points representing country capitals or centroids.

By accepting all occurrences in our database, we would increase the probability to include erroneous records in our analyses, which can lead to an overestimation of species’ ranges. Conversely, applying a given threshold to code a taxon as occurring in an area could lead to the opposite result: under-estimation of species’ ranges if the distribution of the species is poorly known or if some regions are less well sampled than others, which is often the case for Neotropical taxa. Some Neotropical regions accumulate the greatest number of missing or undescribed species, and the lowest biodiversity monitoring efforts (Kier et al. 2005 – J. Biogeogr.; Pimm and Joppa 2015 – Ann. Missouri Bot. Gard.). For example, the southern section of the Amazon basin and northern Colombia, are presumably the most species-rich of all data gaps in the world for plants (Frodin 2001 Cambridge University Press; Perez-Escobar et al. 2022 – TIPS). We considered that no methodological option came without problems.

Given the limited knowledge on the distribution of many Neotropical species, we chose to follow the first approach. However, in order to minimize the impact of this choice, we manually inspected the distribution of a number of records, especially when species were assigned to more than >2 areas. Specifically, we surveyed the distribution of 4,477 species of the 12,512 species included in our dataset, checking for completeness and accuracy with reference to different databases (AmphibiaWeb 2018, Uetz et al. 2018, GBIF.org 2018, IUCN 2018, etc.). Most species in our dataset occurred in a single WWF ecoregion (7,334; please see the table below), reflecting a well-known pattern in the Neotropics of a large proportion of assemblages of endemic species with small ranges (Pimm et al. 2014 – Science). Only 5,178 species occur in more than 1 area. We increased the surveying effort with the number of areas; for comparison, we checked the distribution of 26% of the single area species, and almost 50% of the species with distributions in more than 1 area (from 35% of the species occurring in 2 areas to 100% of the species occurring in 13 areas). We are painfully aware that even with this effort it is still possible that there may be errors in our database, but we still hope that general patterns could be extracted from this data.

Author response table 1 shows the number (Nb.) of species assigned to 1 to 13 WWF ecoregions, and the proportion of species for which we checked their distribution.

**Author response table 1. sa2table1:** 

Nb. WWF areas	Nb. species	Nb. species checked	% species checked
1	7,334	1,918	26.15
2	2,609	922	35.34
3	1,339	529	39.51
4	687	370	53.86
5	307	215	70.03
6	218	184	84.40
7	127	103	81.10
8	111	91	81.98
9	77	67	87.01
10	46	35	76.09
11	32	30	93.75
12	11	11	100
13	6	6	100

L685-687: Was the 60% criterion applied above used here to determine whether clades were scored as mixed, lowland, montane, or highland?

Unfortunately, we were unable to follow the same quantitative classification criteria for coding the main elevation of each clade in our dataset as done for their distribution. Many of the 12,512 species in our 150 phylogenies lacked accurate altitudinal data, as GBIF georeferenced records often come without this information. For example, we estimated that only 40% of the mammal occurrences downloaded in our dataset included elevation data; 34% of the plants. Surprisingly, occurrence data for birds only included associated elevation information in 4% of cases. We therefore considered the altitudinal range of each species could be incompletely described based on the sparse occurrence data available for many species in our dataset, and chose to rely on the descriptions in the literature, keeping in mind that groups described with most of their species in a given elevation would be assigned to this elevation (mimicking the 60% criterion used for the distribution above). We have clarified our approach in the revised version as follows: “We additionally classified clades according to the main elevational range of their constituent species following literature descriptions rather than a purely quantitative approach as for the distribution above, because GBIF records in our dataset often came without associated altitude data (<30%):”

L699-700: Doesn't this show that there is NOT phylogenetic signal (as it is written, it states that any trait contains phylogenetic signal, even though the K values are low and non-significant).

Thanks for noticing this. As suggested, it is in fact the opposite. We have corrected this sentence by replacing “any” by “no”.

[Editors’ note: The authors appealed the decision. What follows is the authors’ response to the second round of review.]

Both Reviewers and we (editors) valued the work done in the revision, but further comments raised by Reviewer 1 further weakened the validity of your approach. Specifically, the lack of a controlled methodology (by confirmatory or accuracy-testing simulations) made it difficult to evaluate if any strong bias exists in your results, or if your datasets have enough power to sustain your claims.Reviewer #1 (Recommendations for the authors):I previously reviewed this manuscript in December of 2021. I had several concerns. One was that the main method used to estimate diversification over time might be problematic.

We thank you for your insightful comments and review. We apologize for not being clear enough in our previous responses to those comments. We have now provided additional information and data, and have included a new section in the manuscript discussing in detail the limitations of the study. We hope that the responses given below and corresponding changes in the manuscript will help clarify these issues.

As previously articulated, we feel this study should not be penalized by the current lack of scientific consensus on a topic that remain central to modern studies in evolutionary biology: the estimation of diversification rates based on molecular phylogenies and time-variable homogeneous birth-death ‘BD’ models. Debate on this subject is intense (Louca and Pennell 2020; Helmstetter et al. 2022; Morlon et al. 2022; Höhna et al. 2022) and is unlikely to be solved in the short term. Our paper does not neglect the debate. By comparing our results with pulled diversification models (supposed to correct for the identifiability issue raised by recent studies associated with traditional diversification rates) as far as these two estimates could be compared, and discussing the limitations of our results, we would argue that we have done our very best efforts to present the limits and extension of our conclusions. We also think that *eLife* is a key journal for publishing results that bring discussion further, like those in this manuscript, rather than avoid topics for which consensus has not yet been reached.

Another was that there was little comparison among different regions to understand patterns of diversity among regions in the Neotropics. I appreciate that the authors have made efforts to address these concerns, but I think that these problems remain somewhat problematic.

We have addressed this comment below, and hopefully you will find this time our answers more satisfactory. However, we think that our points of view and preferred methodology may differ on how biogeographic regions should be delimited and diversification rates analyzed. We think such differential preferences are part of a healthy scientific arena and should be allowed expression, since none of them present logical or obvious flaws.

First, despite what the authors claim, I think the main approach used has not really been thoroughly tested with simulations. The authors claim that it has been, specifically by Morlon et al. (2011), Lewitus and Morlon (2018), and Condamine et al. (2019). The paper by Morlon et al. (2011) contains simulations, but (as far as I know) those simulations do not actually address the ability of the method to accurately choose among different models of diversification over time.

We respectfully disagree on this point. We acknowledge that not all possible simulation scenarios have been performed (there are simply too many cases). But compared to other birth-death models, the RPANDA models have been thoroughly tested with simulations. Time-dependent birthdeath models are not only able to recover (1) the parameter values (including negative net diversification rates, Morlon et al. 2011), but these models are also able to (2) correctly detect shifts of diversification (with regularization techniques as proposed in Morlon et al. 2022), (3) recover the good diversification model, and (4) infer the correct paleodiversity dynamic by taking into account clade, clade diversity and rates (Mazet et al. 2022: https://www.biorxiv.org/content/10.1101/2022.05.10.490920v1).

The paper by Condamine et al. (2019) does not contain simulations at all.

We agree that this study does not include true simulations and have therefore excluded this citation from references about simulations, and refer instead to the Lewitus and Morlon (2018)'s paper which did so (see below our response to your comment related to this paper). In the Condamine et al. (2019)'s paper, the authors have shown that the specificities of the environmental curve matter for recovering a statistical support for a temperature-dependent model.

The paper by Lewitus and Morlon (2018) contains simulations, but primarily compares the ability of the method to distinguish time-dependent and temperature-dependent models (their Figure 3). But it is unclear if these two classes of models can be accurately distinguished from cases where the true model is a constant-rate model. Furthermore, looking at Figure 3 of Lewitus and Morlon (2018) it can be seen that this method often selects a constant-rate model when the true model is temperature dependent, if the effect of temperature dependence on diversification over time is not strong enough. The authors include in this list only papers co-authored by the developer of the method, and not other papers that found the method to be more problematic.

We acknowledge that this method has limitations (as all methods do). Over the last decade, the RPANDA package has incorporated several birth-death models that are now made to be compared in a common statistical framework, which is not the case if we perform other birth-death models like BAMM, RevBayes or SSE-based models. These latter models will be useful to address other questions like the role of traits on diversification or the presence of rate heterogeneity across the studied phylogenies. As we focus on drivers of diversification, there are not many models we can use to test the hypotheses we proposed in the Introduction. Also, we aim at a clear story and do not want to perform a battery of analyses that would make the study long and more confusing for the readers.

We think it's an important point that time-dependent and temperature-dependent models are distinguishable; it’s not the case between time- and diversity-dependent models (Pannetier et al. 2021). Importantly, temperature-dependent models are also able to accurately recover simulated parameter values (even for the extinction rate). According to simulations, the model selection tends to be sensitive when the dependency parameter [α] ranges between -0.1 and 0.1. This means that we are conservative when we conclude that temperature-dependent models are estimated as best fitting.

We would like to point out that we have calculated the proportion of phylogenies whose diversification rates are best explained by a model with constant, time-dependent, temperaturedependent, or uplift-dependent diversification based on (1) the best fitting model, and (2) the secondbest model (Figure-5—figure supplement 1). If the constant-rate model overfits temperaturedependent trees, then we would expect that the temperature-dependent models are the second bestfit model in most cases. We find it's not the case in our analyses. Of the 76 clades that have a constantrate model as the best fit, 50% (38/76) have temperature-dependent models as the second best fit, 40% (30/76) have time-dependent models, and 10% (8/76) have Andean-dependent models. Please see our new Figure-5-source-data-2. This suggests that there is no major bias in our model selection. These results can even suggest there would be a similar bias towards time-dependent models. Furthermore, when evaluating the α values of those clades which are best fit by a constant-rate model and second best fit by a temperature-dependent model, we find that only 26% (10/38) of the temperature models have α values ranging between -0.1 and 0.1, which according to simulations is the value range where model selection tends to be sensitive. (Please see Author response image 3 and our new Figure-5-source-data-2). This number gets reduced to 6% when considering the whole sample size (10/150), meaning that only 6% of the trees in our dataset are susceptible to suffer from this potential bias (i.e. select a constantrate models when the true model is a temperature-dependent model).

**Author response image 3. sa2fig3:** Α value of the clades which are best fit by a constant-rate model and second best fit by a temperature-dependent model. In red those values ranging between -0.1 and 0.1. Eleven trees have constant speciation (α does not vary) and thus they are not represented here.

We have added this information (model sensitivity and evaluation of our data) to a new section ("Limitations and perspectives") in the Discussion. We now discuss these limitations of the RPANDA approach on this section.

It is also notable that the diversification dynamics are not significantly different between plants and animals when using PDR (Table 2). Therefore, not all of the main results are concordant between these two methods.

We may not have been sufficiently clear in reporting these results. In fact, Table 2 does show that pulled diversification rates (PDR) are significantly different between plants and animals (*P* < 0.00), in agreement with results based on traditional BD models. Diversification trajectories, derived from these rates, are not significantly different in terms of proportion of clades experiencing expanding vs. declining speciation trends across taxonomic groups. Unfortunately, inferences on speciation trends are not sufficient to derive net diversification dynamics and thus to support/reject results based on BD rates. Extinction cannot be inferred based on PDR, only speciation (Louca and Pennel 2020), but estimates of extinction are needed to derive net diversification dynamics (net diversification = speciation minus extinction), as we did base on BD models. For example, we cannot rule out that the decreases in speciation in plants based on PDR (Figure 3) were accompanied by larger decreases in extinction. Under this scenario, net diversification rates could still increase through time, and we could still recover similar expanding diversification dynamics for plants as we found based on BD models. Thus, direct comparisons between traditional BD and PDR models were not possible in this case.

We explain in the Discussion that results from PDR and BD are not directly comparable on this point, but we realize this might have been confusing or unclear in the previous version. We have therefore clarified it further in the text. It now reads: “The study of PDR did not help to confirm/reject these conclusions. Rates from PDR models are significantly different between plants and animals (p<0.00), in agreement with results based on traditional models (Table 2). Diversification trajectories derived from these rates are not different, with plants and animals showing an equivalent fraction of phylogenies showing a decay of speciation (Figure 3b). Since extinction dynamics cannot be derived from PDR models, we do not know if speciation slowdowns detected in plants based on PDR were accompanied by larger extinction declines (Figure 3). Thus, we cannot rule out the scenario of expanding dynamics (Sc. 1, 2) for plants found based on traditional birth-death models.”

I found the comparison among biogeographic regions to be unsatisfying overall, because the regions are extremely coarse grained and each incorporates so many different regions.

We understand why the reviewer finds the biogeographic regions identified here as being broad in comparison with traditional subdivisions of the Neotropics. We also agree with the principle of using the maximum possible resolution afforded by data availability, and agree that "finer" regions would be desirable if our evolutionary units of analysis were species – as has been done in some previous studies comparable to ours. In contrast, our study is novel in using clades (lineages that radiated in the Neotropics) as evolutionary units for the spatial analyses.

While broad biogeographic regions would not be meaningful when working at the species level, they are adequate when working at the clade-level scale. In fact, we initially tried a finer definition of Neotropical regions, but it did not work for our data. Figure-6-source-data-1 shows that finer regions, such as ecoregions (Olson, 2001), do not represent patterns of endemism for our clades, i.e., most of the clades have species distributed in most ecoregions, and almost none is endemic to one ecoregion. Another important point against fine-scale biogeographic regions is that the origin of many of the traditional Neotropical subdivisions postdates the origin of many of the studied clades.

For example, the Cerrado formed in the late Miocene (Simon et al., 2009), and the Chocó in the Pliocene-Pleistocene (Pérez-Escobar et al., 2019), which stresses the conclusion that finer bioregions were not yet formed when most clades originated (Figure 2).

We would like to stress that the definition of our biogeographic regions is the result of a quantitative analysis, rather than resulting from any *a priori* delimitation based on expert’s knowledge. Our regions derive from the analysis of patterns of endemicity in our sample of 150 Neotropical clades. Importantly, these regions are overall congruent with the biogeographic sub-regionalization of the Neotropics proposed in previous influential studies (Morrone 2017, 2022).

We believe that the use of clades (rather than species) as evolutionary units in our biogeographic comparisons is a novelty of our study and has several advantages. We were able to compare diversification trends through time (i.e. constant, expanding, declining) across regions, and not just present-day diversification rates as in previous studies (Harvey et al., 2020 – Science; Quintero and Jetz 2018 – Nature; Smith et al., 2014 – Nature). The later approach is limited because when rates vary through time, diversification could be higher in one region than in another without providing information on the underlying diversification dynamic through time. We think the existence of regional arenas of diversification in the Neotropics at the clade level and at deep time scales is a key finding of this study. We have extended the discussion to explain this: “The use of clades (rather than species) as evolutionary units in our biogeographic comparisons is original, and allowed us to compare diversification trends through time (i.e. constant, expanding, declining) across regions, and not just present-day diversification rates, as in different comparable studies (e.g., Harvey et al., 2020; Quintero and Jetz 2018; Smith et al., 2014). Present-day diversification rates are structured geographically in the Neotropics (Harvey 2020, Jetz 2012, Quintero and Jetz 2018; Rangel 2018), but our study shows that present diversification do not represent long-term evolutionary dynamics. The lack of a clear geographic structure of long-term diversification suggests that the evolutionary forces driving diversity in the Neotropics acted at a continental scale when evaluated over tens of millions of years. Evolutionary time and extinction could have eventually acted as levelling agents of diversification across the Neotropics over time.”

Furthermore, the comparison of 97 clades in one region (Mesoamerica+Amazonia+Andes and others) vs. only 10 clades in another (southern South America) also seems unsatisfying.

We agree with the reviewer. Our conclusions derive from the study of a fraction of the Neotropical diversity, where tropical rainforest lineages from the broad “pan-Amazonian” region are most abundant. Although sample size in our study is large (150 observations), some categories of these variables are poorly represented, which might limit the performance of some statistical tests. For instance, there are 97 phylogenies distributed in the biogeographic cluster 1, while only 10 in cluster 2, 4 in cluster 3. Note that there are other clades (39) containing species on poorly represented regions that fall in the “mixed” category, as they share species with different areas. We argue, however, that our sampling does include representatives from all the main regions in the Neotropics. Yet, we did not identify a common diversification trajectory among the fewer clades distributed on poorly represented regions (e.g. southern South America clades experienced all gradual, exponential, and declining dynamics, as did the clades from other regions; Figure 6). Furthermore, it is reasonable to assume that our sampling reflects a fair proportion of species’ distributions in the Neotropics considering the extension of these regions in the Neotropics, and the representativeness of our dataset; at least for tetrapods, it includes ~60% of all described species. We did not choose our clades *a priori* for belonging to one area or the other, and our sampling probably represents the distribution of species on these regions. Furthermore, the hypothesis of comparable diversification gains support when comparing raw diversification rate estimates (Figure 6d,e), and not just their derived species richness trends (Sc. 1–4; Figure 6f,g). Still, future studies with larger sample sizes per region will be needed to clarify the generality of our results. We have expanded on this issue in the new section ("Limitations and perspectives") in the Discussion.

I appreciate that the authors describe their four scenarios explicitly in the Introduction, but I do not think that knowing the frequency of these four scenarios is "required to understand the origin and maintenance of Neotropical diversity" (line 112).

It is true that describing the frequency of these four scenarios on the dynamics of Neotropical diversity over the Cenozoic will not resolve "the origin and maintenance of Neotropical diversity", but we think that this information can help to shed light on it. We have rephrased this statement in the Introduction, which now reads: “Yet, such an assessment would contribute to understand the origin and maintenance of Neotropical diversity.”

Also, I do not see why the accumulation of species through "pulses" corresponds to the exponential expansion scenario. Should not constant speciation and low, constant extinction lead to an exponential increase in diversity over time? Even in Figure 1, I do not see the pulses in Scenario 2. A "pulse" implies a pattern that is discontinuous or episodic over time.

The reviewer is right. The term “pulses of speciation” refers to episodic discontinuous events and does not accurately describe the exponential increase of diversity described in Scenario 2. We have corrected the text, which now reads: “*An exponential increase in diversity model asserts that species richness accumulated faster towards the present*”. Note that a scenario of constant speciation and low constant extinction will not lead to an exponential increase in diversity over time, but to a linear increase (see scenario on Figure 1b).

The finding that diversification patterns are different between plants and animals is interesting, but not entirely novel (see for example, Hernandez-Hernandez et al. 2021; Biological Reviews on the faster rates of diversification in plants vs. animals). It is also notable (again) that the diversification dynamics over time are not significantly different between plants and animals when using PDR, even if the rates are.

Thank you for this relevant reference. We have included this reference in the discussion. Now reads: “Net diversification rates were also higher in plants (Figure 8), in agreement with previous studies (Hernandez-Hernandez et al., 2021)”. Regarding the apparent contradiction between the PDR and BD results, please see our previous reply.

Finally, I am not sure how relevant these results are to future climate change, as is stated in the Abstract. It seems like this idea is not addressed in the paper itself.

Thank you for this comment. We have rephrased the sentence as follows: “These opposite evolutionary patterns may reflect different capacities for plants and tetrapods to cope with past climate changes”.

Specific comments:Lines 285 to 287: This needs to be rewritten since the results seem to contradict the conclusions.

Thank you for noticing this mistake. Following the suggestion of Reviewer 3 we have corrected the sentence as follows: “we find that no continuous or multi-categorical trait shows a phylogenetic signal”.

Lines 340-342. The study cited focused on BiSSE-type models, not the approach used here.

The reviewer is right. Simulations in Davis et al. (2013) were conducted using BiSSE-type models. However, sample size limitations are not particular of birth-death models, but to any kind of model. We have changed this sentence as follows: “Alternatively, the power of birth–death models to detect rate variation decreases with the number of species in a phylogeny, as shown with different diversification approaches (Davis et al. 2013: Lewitus and Morlon 2018; Burin et al. 2019)”.

Line 374. Change "plant phylogenies are significantly worst sampled than those of most other tetrapods" to "plant phylogenies are significantly less sampled than are tetrapod phylogenies".

Thanks! Corrected.

Lines 415 to 416. How can there be microevolutionary studies of diversification?

The reviewer is right. We have reworded this paragraph and delete this sentence.

Lines 538 to 540. The paper by Kreft and Jetz does not address diversification, only present-day species richness.

Thanks for noticing this misquotation. We have removed this reference and replaced it by: Jetz et al., 2012; Harvey et al., 2020; Quintero and Jetz 2018; Rangel et al., 2018.

Lines 554 to 555. The "diverse range of diversification methods compared here" is actually only two diversification methods, by my count.

We have corrected this sentence. It now reads: “and the diversification models compared here”.

Line 563. What "substantial proportion of Neotropical diversity" is accounted for by these models?

The proportion of models accounted by saturated and declining models is 30%. If the reviewer agrees, we prefer not to include this information in the conclusions, as we do not think it is the appropriate place to put numbers. Besides, it is already presented in the abstract, results and discussion.

[Editors’ note: what follows is the authors’ response to the third round of review.]

In this revised version, we stress that you must carefully address the following: (1) Please be more cautious about reporting the constraints of your chosen methods, (2) Please dilute the claim that you are resolving South American diversification, (3) Please address the several aspects of the most recent reviews which were not addressed in your appeal letter, including (but not limited to): your reliance on one specific method to calculate diversification rates, an apparent lack of simulations to support your chosen method, the coarse-grained resolution of the study and imbalance in clade numbers investigated in different regions, the request for a clearer and more critical projection of scenarios resulting from the proposed speciation patterns (do pulses necessarily lead to exponential expansion), and being careful to base conclusions about diversification patterns on dynamics and rates.

Thank you very much for considering our appeal letter and for allowing us to submit a revised manuscript and a response to the reviewer's comments. We are also grateful for the suggestions of the three reviewers whose comments contributed to improve the paper. We have carefully addressed all the comments raised by the reviewers — please see our detailed answers below in blue font.

In summary, we agree with the suggestions put forward and have therefore toned down throughout the main text our claims that the results of our analyses are resolving the origin of the extraordinary South American diversity. Of course, such claim would be inappropriate, as our results can only contribute to our overall understanding of this complex and multifaceted topic. We have also tried to clarify what has and has not been done regarding simulations and birth-death modeling (please see our answer to reviewer 1 and our changes in the main text). We also discuss how, compared to other birth-death models, the RPANDA models have been thoroughly tested using simulations.

We acknowledge that all possible simulation scenarios have not been performed – there are simply too many possibilities. Previous studies, cited in the revised text, show that these models are not only able to recover the parameter values – including negative net diversification rates, which are often underestimated by models such as BAMM – but are also able to detect shifts of diversification, as well as the best-fitting diversification model. According to previous simulations (e.g., Lewitus and Morlon 2018), model selection tends to be sensitive to some particular scenarios. We now explicitly discuss such scenarios, and quantify the frequency with which they might potentially occur in our data (ca. 6% of our trees; please see our new Figure-5-source-data-2). This evaluation indicates that potential methodological biases are not significantly affecting our study, nor would have an effect on our conclusions.

We have now worked further to improve and clarify the reliability and limitations of our study by adding a specific section ("Limitations and perspectives") in the Discussion, in which we not only explain why birth-death models in RPANDA and diversification parameters are demonstrably identifiable, but also highlight that our results are based on traditional diversification rates and on the Pulled Diversification Rates method (which corrects for the identifiability issue raised by studies associated with traditional diversification rates).

Regarding the “coarse” definition of biogeographic areas raised by the reviewer, we have very carefully considered this feedback. We agree with the principle of using the maximum possible resolution afforded by data availability, and agree that "finer" regions would be desirable if our evolutionary units of analysis were species – as has been done in some previous studies comparable to ours. In contrast, our study is novel in using clades (lineages that radiated in the Neotropics) as evolutionary units for the spatial analyses. This means that finer areas cannot be used to categorize clade distributions for the simple reason that some of the finer regions used in previous studies conducted at the species level did not exist at the origin (and during the evolutionary history) of most of the clades.

As mentioned in the previous round of reviews, we actually attempted to apply finer regions and quickly realized that they did not produce biologically and biogeographically meaningful results. As shown in our Figure-6-source-data-1, finer regions, such as ecoregions (Olson, 2001), do not represent patterns of endemism for our clades, i.e., most of the clades have species distributed in most ecoregions, and almost none is endemic to one ecoregion. To analyze clade endemism and perform biogeographic comparisons of diversification at the clade level, we therefore needed to identify broader regions. We have done this following a quantitative analysis, which the reviewer has not found to be problematic, and not based on any *a priori* subjective decision. Moreover, our regions are overall congruent with the biogeographic sub-regionalization of the Neotropics proposed in influential works by Morrone (2017, 2022), which will make the results of our study more easily comparable with previous studies.

Although using clades (rather than species) as evolutionary units requires a larger spatial resolution for the biogeographic analyses, we think that our approach represents a major novelty of the study, bringing several advantages. Most importantly, we were able to compare diversification trends through time (i.e. constant, expanding, declining) across regions, and not just present-day diversification rates, as in comparable previous studies (e.g., Harvey et al., 2020 – Science; Quintero and Jetz 2018 – Nature; Smith et al., 2014 – Nature). A species-based approach is limited in scope because when rates vary through time, present diversification could be higher in one region than in another, without providing information on the underlying temporal dynamic. We think the existence of regional arenas of evolution in the Neotropics at the clade level and at large time scales is a key finding of this study and it should stimulate debate and further research. We have tried to clarify this in the reply to the reviewer and in the revised manuscript.